# THE CANARY'S ECHO: AUDITING PRIVACY RISKS OF LLM-GENERATED SYNTHETIC TEXT

## ABSTRACT

How much information about training examples can be gleaned from synthetic data generated by Large Language Models (LLMs)? Overlooking the subtleties of information flow in synthetic data generation pipelines can lead to a false sense of privacy. In this paper, we investigate the design of membership inference attacks that target data used to fine-tune pre-trained LLMs that are then used to synthesize data, particularly when the adversary does not have access to the fine-tuned model but only to a synthetic data corpus. We demonstrate that canaries crafted to maximize vulnerability to attacks that have access to the model are sub-optimal for auditing privacy risks when only synthetic data is released. This is because such out-of-distribution canaries have limited influence on the model's output when prompted to generate useful, in-distribution synthetic data, which drastically reduces their vulnerability. To tackle this problem, we leverage the mechanics of auto-regressive models to design canaries that leave detectable traces in synthetic data. Our approach greatly enhances the power of membership inference attacks, providing a better assessment of the privacy risks of releasing synthetic data generated by LLMs.

## 1 INTRODUCTION

Large Language Models (LLMs) can generate synthetic data that mimics human-written content through domain-specific prompts. Besides their impressive fluency, LLMs are known to memorize parts of their training data (Carlini et al., 2023) and can regurgitate exact phrases, sentences, or even longer passages when prompted adversarially (Zanella-Béguelin et al., 2020; Carlini et al., 2021; Nasr et al., 2023). This raises serious privacy concerns about unintended information leakage through synthetically generated text. In this paper, we address the critical question: how much information about real data leaks through text synthetically generated from it using LLMs?

Prior methods to audit privacy risks insert highly vulnerable out-of-distribution examples, known as *canaries* (Carlini et al., 2019), into the training data and test whether they can be identified using membership inference attacks (MIAs) (Shokri et al., 2017). Various MIAs have been proposed, typically assuming that the attacker has access to the model or its output logits (Carlini et al., 2019; Shi et al., 2024). In the context of LLMs, MIAs often rely on analyzing the model's behavior when prompted with inputs related to the canaries (Carlini et al., 2021; Chang et al., 2024; Shi et al., 2024). However, similar investigations are lacking in scenarios where LLMs are used to generate synthetic data and only this synthetic data is made available.

**Contributions** In this work, we study–for the first time–the factors that influence leakage of information about a data corpus from synthetic data generated from it using LLMs. First, we introduce data-based attacks that only have access to synthetic data, not the model used to generate it, and therefore cannot probe it with adversarial prompts nor compute losses or other statistics used in model-based attacks (Ye et al., 2022; Carlini et al., 2022a). We propose approximating membership likelihood using either a model trained on the

synthetic data or the target example similarity to its closest synthetic data examples. We design our attacks adapting pairwise likelihood ratio tests as in RMIA (Zarifzadeh et al., 2024) and evaluate our attacks on labeled datasets: SST-2 (Socher et al., 2013) and AG News (Zhang et al., 2015). Our results show that MIAs leveraging only synthetic data achieve AUC scores of 0.71 for SST-2 and 0.66 for AG News, largely outperforming a random guess baseline. This suggests that synthetic text can leak significant information about the real data used to generate it.

Second, we use the attacks we introduce to quantify the gap in performance between data- and model-based attacks. We do so in an auditing scenario, designing adversarial canaries and controlling leakage by varying the number of times a canary occurs in the fine-tuning dataset. Experimentally, we find a sizable gap when comparing attacks adapted to the idiosyncrasies of each setting: a canary would need to occur $8\times$ more often to be as vulnerable against a data-based attack as it is against a model-based attack (see Fig. 1).

Third, we discover that canaries designed for model-based attacks fall short when auditing privacy risks of synthetic text. Indeed, privacy auditing of LLMs through model-based MIAs relies on rare, out-of-distribution sequences of high perplexity (Carlini et al., 2019; Stock et al., 2022; Wei et al., 2024; Meeus et al., 2024c). We confirm that model-based MIAs improve as canary perplexity increases. In sharp contrast, we find that high perplexity sequences, although distinctly memorized by the target model, are less likely to be echoed in synthetic data generated by the target model. Therefore, as a canary perplexity increases, the canary influence on synthetic data decreases, making its membership less detectable from synthetic data (see Figure 2). We show that low-perplexity, and even in-distribution canaries, while suboptimal for model-based attacks, are more adequate canaries in data-based attacks.

Lastly, we propose an alternative canary design tailored for data-based attacks based on the following observations: (i) in-distribution canaries aligned with the domain-specific prompt can influence the generated output, and (ii) memorization is more likely when canaries contain sub-sequences with high perplexity. We construct canaries starting with an in-distribution prefix of length $F$, transitioning into an out-of-distribution suffix, increasing the likelihood that the model memorizes them and that they influence synthetic data. We show that, for fixed overall canary perplexity, the true positive rate (TPR) of attacks increases by up to $2\times$ by increasing the length of the in-distribution prefix (see Fig. 1). Moreover, we find the MIA performance (both AUC and TPR at low FPR) for canaries with in-distribution prefix and out-of-distribution suffix ($0 < F < \text{max}$) to improve upon both entirely in-distribution canaries ($F = \text{max}$) and out-of-distribution canaries ($F = 0$), for both datasets.

In terms of real-world applications, the novel MIAs and canary design that we propose can be used to audit privacy risks of synthetic text. Auditing establishes a lower bound on privacy risks, which is useful to take informed decisions about releasing synthetic data in sensitive applications (e.g., patient-clinician conversations, customer assistance chats). These lower bounds complement upper bounds on privacy risks from methods that synthesize text with provable guarantees, notably, differential privacy (DP). Auditing can not only detect violations of DP guarantees stemming from flawed analyses, implementation bugs, or incorrect assumptions, but also allows for less conservative decisions based on the performance of MIAs matching the threat model of releasing synthetic data. In contrast, for data synthesized from models fine-tuned with DP guarantees, DP bounds the risk of both model- and data-based attacks and hence does not account for the inherent gap in attacker capabilities that we observe.

## 2 BACKGROUND AND PROBLEM STATEMENT

**Synthetic text generation** We consider a private dataset $D = \{x_i = (s_i, \ell_i)\}_{i=1}^{N}$ of labelled text records where $s_i$ represents a sequence of tokens (e.g. a product review) and $\ell_i$ is a class label (e.g. the review sentiment). A synthetic data generation mechanism is a probabilistic procedure mapping $D$ to a synthetic dataset $\widetilde{D} = \{\widetilde{x}_i = (\widetilde{s}_i, \widetilde{\ell}_i)\}_{i=1}^{\widetilde{N}}$ with a desired label set $\{\ell_i\}_{i=1}^{\widetilde{N}}$. Unless stated otherwise, we consider

$N = \widetilde{N}$. The synthetic dataset $\widetilde{D}$ should preserve the *utility* of the private dataset $D$, i.e., it should preserve as many statistics of $D$ that are useful for downstream analyses as possible. In addition, a synthetic data generation mechanism should preserve the *privacy* of records in $D$, i.e. it should not leak sensitive information from the private records into the synthetic records. The utility of a synthetic dataset can be measured by the gap between the utility achieved by $\widetilde{D}$ and $D$ in downstream applications. The fact that synthetic data is not *directly* traceable to original data records does not mean that it is free from privacy risks. On the contrary, the design of a synthetic data generation mechanism determines how much information from $D$ leaks into $\widetilde{D}$ and should be carefully considered. Indeed, several approaches have been proposed to generate synthetic data with formal privacy guarantees (Kim et al., 2021; Tang et al., 2024; Wu et al., 2024; Xie et al., 2024). We focus on privacy risks of text generated by a pre-trained LLM fine-tuned on a private dataset $D$ (Yue et al., 2023; Mattern et al., 2022; Kurakin et al., 2023). Specifically, we fine-tune an LLM $\theta_0$ on records $(s_i, \ell_i) \in D$ to minimize the loss in completing $s_i$ conditioned on a prompt template $\mathsf{p}(\ell_i)$, obtaining $\theta$. We then query $\theta$ using the same prompt template to build a synthetic dataset $\widetilde{D}$ matching a given label distribution.

**Membership inference attacks**    MIAs (Shokri et al., 2017) provide a meaningful measure for quantifying the privacy risks of machine learning (ML) models, due to its simplicity but also due to the fact that protection against MIAs implies protection against more devastating attacks such as attribute inference and data reconstruction (Salem et al., 2023). In a MIA on a target model $\theta$, an adversary aims to infer whether a target record is present in the training dataset of $\theta$. Different variants constrain the adversary's access to the model, ranging from full access to model parameters (Nasr et al., 2019) to query access (Zarifzadeh et al., 2024). In our setting, we consider adversaries that observe the output logits on inputs of their choosing of a model $\theta$ fine-tuned on a private dataset $D$. We naturally extend the concept of MIAs to synthetic data generation mechanisms by considering adversaries that only observe a synthetic dataset $\widetilde{D}$ generated from $D$.

**Privacy auditing using canaries**    A common method used to audit the privacy risks of ML models is to evaluate the MIA vulnerability of canaries, i.e., artificial worst-case records inserted in otherwise natural datasets. This method can also be employed to derive statistical lower bounds on the differential privacy guarantees of the training pipeline (Jagielski et al., 2020; Zanella-Béguelin et al., 2023). Records crafted to be out-of-distribution w.r.t. the underlying data distribution of $D$ give a good approximation to the worst-case (Carlini et al., 2019; Meeus et al., 2024c). Canaries can take a range of forms, such as text containing sensitive information (Carlini et al., 2019) and random (Wei et al., 2024) or synthetically generated sequences (Meeus et al., 2024c). Prior work identified that longer sequences, repeated more often (Carlini et al., 2023; Kandpal et al., 2022), and with higher perplexity (Meeus et al., 2024c) are better memorized during training and hence are more vulnerable to model-based MIAs. We study multiple types of canaries and compare their vulnerability against model- and synthetic data-based MIAs. We consider a set of canaries $\{\hat{x}_i = (\hat{s}_i, \hat{\ell}_i)\}_{i=1}^{\hat{N}}$, each crafted adversarially and inserted with probability ½ into the private dataset $D$. The resulting dataset is then fed to a synthetic data generation mechanism. We finally consider each canary $\hat{x}_i$ as the target record of a MIA to estimate the privacy risk of the generation mechanism (or the underlying fine-tuned model).

**Threat model**    We consider an adversary $\mathcal{A}$ who aims to infer whether a canary $\hat{x}$ was included in the private dataset $D$ used to synthesize a dataset $\widetilde{D}$. We distinguish between two threat models: (i) an adversary $\mathcal{A}$ with query-access to output logits of a target model $\theta$ fine-tuned on $D$, and (ii) an adversary $\widetilde{\mathcal{A}}$ with only access to the synthetic dataset $\widetilde{D}$. To the best of our knowledge, for text data this latter threat model has not been studied extensively in the literature. In contrast, the privacy risks of releasing synthetic tabular data are much better understood (Stadler et al., 2022; Yale et al., 2019; Hyeong et al., 2022; Zhang et al., 2022). Algorithm 1 shows the generic membership inference experiment encompassing both model- and data-based attacks, selected by the synthetic flag. The adversary is represented by a stateful procedure $\mathcal{A}$,

---

**Algorithm 1** Membership inference against an LLM-based synthetic text generator

---

1: **Input**: Fine-tuning algorithm $\mathcal{T}$, pre-trained model $\theta_0$, private dataset $D = \{x_i = (s_i, \ell_i)\}_{i=1}^N$, labels $\{\widetilde{\ell_i}\}_{i=1}^{\widetilde{N}}$, prompt template $\mathsf{p}(\cdot)$, canary repetitions $n_{\text{rep}}$, sampling method $\mathsf{sample}$, adversary $\mathcal{A}$
2: **Output**: Membership score $\beta$
3: $\hat{x} \leftarrow \mathcal{A}(\mathcal{T}, \theta_0, D, \{\widetilde{\ell_i}\}_{i=1}^{\widetilde{N}}, \mathsf{p}(\cdot))$          ▷ *Adversarially craft a canary (see Sec. 3.2)*
4: $b \sim \{0, 1\}$                                   ▷ *Flip a fair coin*
5: **if** b = 1 **then**
6:     $\theta \leftarrow \mathcal{T}(\theta_0, D \cup \{\hat{x}\}^{n_{\text{rep}}})$         ▷ *Fine-tune $\theta_0$ with canary repeated $n_{rep}$ times*
7: **else**
8:     $\theta \leftarrow \mathcal{T}(\theta_0, D)$                   ▷ *Fine-tune $\theta_0$ without canary*
9: **for** $i = 1 \ldots \widetilde{N}$ **do**
10:     $\widetilde{s}_i \sim \mathsf{sample}(\theta(\mathsf{p}(\widetilde{\ell}_i)))$       ▷ *Sample synthetic records using prompt template*
11: $\widetilde{D} \leftarrow \left\{ (\widetilde{s}_i, \widetilde{\ell}_i) \right\}_{i=1}^{\widetilde{N}}$
12: **if** $\mathsf{synthetic}$ **then**                       ▷ *Compute membership score $\beta$ of $\hat{x}$*
13:     $\beta \leftarrow \mathcal{A}(\widetilde{D}, \hat{x})$           ▷ *See Sec. 3.1.2 and algorithms in Appendix A*
14: **else**
15:     $\beta \leftarrow \mathcal{A}(\theta, \hat{x})$                     ▷ *See Sec. 3.1.1*
16: **return** $\beta$

---

used to craft a canary and compute its membership score. Compared to a standard membership experiment, we consider a fixed private dataset $D$ rather than sampling it, and let the adversary choose the target $\hat{x}$. This is close to the threat model of *unbounded* differential privacy, where the implicit adversary selects two datasets, one obtained from the other by adding one more record, except that in our case the adversary observes but cannot choose the records in $D$. The membership score $\beta$ returned by the adversary can be turned into a binary membership label by choosing an appropriate threshold. We further clarify assumptions made for the adversary in both threat models in Appendix D.

**Problem statement**    We study methods to audit privacy risks associated with releasing synthetic text. Our main goal is to develop an effective data-based adversary $\widetilde{\mathcal{A}}$ in the threat model of Algorithm 1. For this, we explore the design space of canaries to approximate the worst-case, and adapt state-of-the-art methods used to compute membership scores in model-based attacks to the data-based scenario.

## 3 METHODOLOGY

### 3.1 COMPUTING THE MEMBERSHIP SCORE

In Algorithm 1, the adversary computes a membership score $\beta$ indicating their confidence that $\theta$ was trained on $\hat{x}$ (i.e. that $b = 1$). We specify first how to compute a membership signal $\alpha$ for model- and data-based adversaries, and then how we compute $\beta$ from $\alpha$ adapting the RMIA methodology of Zarifzadeh et al. (2024).

### 3.1.1 MEMBERSHIP SIGNAL FOR MODEL-BASED ATTACKS

The larger the target model $\theta$'s probability for canary $\hat{x} = (\hat{s}, \hat{\ell})$, $P_\theta(\hat{s} \mid \mathsf{p}(\hat{\ell}))$, as compared to its probability on reference models, the more likely that the model has seen this record during training. We compute the probability for canary $\hat{x}$ as the product of token-level probabilities for $\hat{s}$ conditioned on the prompt $\mathsf{p}(\hat{\ell})$. Given

a target canary text $\hat{s} = t_1, \ldots, t_n$, we compute $P_\theta(\hat{s} \mid \mathsf{p}(\hat{\ell}))$ as $P_\theta(\hat{x}) = \prod_{j=1}^n P_\theta(t_j \mid \mathsf{p}(\hat{\ell}), t_1, \ldots, t_{j-1})$. We consider this probability as the membership inference signal against a model, i.e. $\alpha = P_\theta(\hat{s} \mid \mathsf{p}(\hat{\ell}))$.

### 3.1.2 MEMBERSHIP SIGNAL FOR DATA-BASED ATTACKS

When the attacker only has access to the generated synthetic data, we need to extract a signal that correlates with membership purely from the synthetic dataset $\widetilde{D}$. We next describe two methods to compute a membership signal $\alpha$ based on $\widetilde{D}$. For more details, refer to their pseudo-code in Appendix A.

**Membership signal using $n$-gram model**  The attacker first fits an $n$-gram model using $\widetilde{D}$ as training corpus. An $n$-gram model computes the probability of the next token $w_j$ in a sequence based solely on the previous $n-1$ tokens (Jurafsky & Martin, 2024). The conditional probability of a token $w_j$ given the previous $n-1$ tokens is estimated from the counts of $n$-grams in the training corpus. Formally,

$$P_{n\text{-gram}}(w_j \mid w_{j-(n-1)}, \ldots, w_{j-1}) = \frac{C(w_{j-(n-1)}, \ldots, w_j) + 1}{C(w_{j-(n-1)}, \ldots, w_{j-1}) + V}, \tag{1}$$

where $C(s)$ is the number of times the sequence $s$ appears in the training corpus and $V$ is the vocabulary size. We use Laplace smoothing to deal with $n$-grams that do not appear in the training corpus, incrementing by 1 the count of every $n$-gram. The probability that the model assigns to a sequence of tokens $s = (w_1, \ldots, w_k)$ can be computed using the chain rule: $P_{n\text{-gram}}(s) = \prod_{j=2}^k P_{n\text{-gram}}(w_j \mid w_{j-(n-1)}, \ldots, w_{j-1})$. With the $n$-gram model fitted on the synthetic dataset, the attacker computes the $n$-gram model probability of the target canary $\hat{x} = (\hat{s}, \hat{\ell})$ as its membership signal, i.e. $\alpha = P_{n\text{-gram}}(\hat{s})$. The intuition here is that if the canary $\hat{x}$ was present in the training data, the generated synthetic data $\widetilde{D}$ will better reflect the patterns of $\hat{s}$, resulting in the $n$-gram model assigning a higher probability to $\hat{s}$ than if it was not present.

**Membership signal using similarity metric**  The attacker computes the similarity between the target canary text $\hat{s}$ and all synthetic sequences $\widetilde{s}_i$ in $\widetilde{D}$ using some similarity metric SIM, i.e. $\sigma_i = \text{SIM}(\hat{s}, \widetilde{s}_i)$ for $i = 1, \ldots, \widetilde{N}$. Next, the attacker identifies the $k$ synthetic sequences with the largest similarity to $\hat{s}$. Let $\sigma_{i(j)}$ denote the $j$-th largest similarity. The membership inference signal is then computed as the mean of the $k$ most similar examples, i.e. $\alpha = \frac{1}{k} \sum_{j=1}^k \sigma_{i(j)}$. The intuition here is that if $\hat{s}$ was part of the training data, the synthetic data $\widetilde{D}$ will likely contain sequences $\widetilde{s}_i$ more similar to $\hat{s}$ than if $\hat{s}$ was not part of the training data, resulting in a larger mean similarity. Various similarity metrics can be used. We consider Jaccard similarity ($\text{SIM}_{\text{Jac}}$), often used to measure string similarity, and cosine similarity between the embeddings of the two sequences, computed using a pre-trained embedding model ($\text{SIM}_{\text{emb}}$).

### 3.1.3 LEVERAGING REFERENCE MODELS TO COMPUTE RMIA SCORES

Reference models, also called *shadow* models, are surrogate models designed to approximate the behavior of a target model. MIAs based on reference models perform better but are more costly to run than MIAs that do not use them, with the additional practical challenge that they require access to data distributed similarly to the training data of the target model (Shokri et al., 2017; Ye et al., 2022). Obtaining multiple reference models in our scenario requires fine-tuning a large number of parameters in an LLM and quickly becomes computationally prohibitive. We use the state-of-the-art RMIA method (Zarifzadeh et al., 2024) to maximize attack performance with a limited number of reference models $M$. Specifically, for the target model $\theta$, we calculate the membership score of a canary $\hat{x}$ using reference models $\{\theta_i'\}_{i=1}^M$ as follows (we present the details on the application of RMIA to our setup in Appendix B):

$$\beta_\theta(\hat{x}) = \frac{\alpha_\theta(\hat{x})}{\frac{1}{M} \sum_{i=1}^M \alpha_{\theta_i'}(\hat{x})}. \tag{2}$$

## 3.2 CANARY GENERATION

Prior work has shown that canaries with high perplexity are more likely to be memorized by language models (Meeus et al., 2024c). High perplexity sequences are less predictable and require the model to encode more specific, non-generalizable details about them. However, high perplexity canaries are not necessarily more susceptible to leakage via synthetic data generation, as they are outliers in the text distribution when conditioned on a given in-distribution prompt. This misalignment with the model's natural generative behavior means that even when memorized, these canaries are unlikely to be reproduced during regular model inference, making them ineffective for detecting memorization of training examples in generated synthetic data.

To address this issue, we take advantage of the greedy nature of popular autoregressive decoding strategies (e.g. beam search, top-$k$ and top-$p$ sampling). We can encourage such decoding strategies to generate text closer to canaries by crafting canaries with a low perplexity prefix. To ensure memorization, we follow established practices and choose a high perplexity suffix. Specifically, we design canaries $\hat{x} = (\hat{s}, \hat{\ell})$, where $\hat{s}$ has an **in-distribution prefix** and an **out-of-distribution suffix**. In practice, we split the original dataset $D$ into a training dataset and a canary source dataset. For each record $x = (s, \ell)$ in the canary source dataset, we design a new canary $\hat{x} = (\hat{s}, \hat{\ell})$. We truncate $s$ to get an in-distribution prefix of length $F$ and generate a suffix using the pre-trained language model $\theta_0$, adjusting the sampling temperature to achieve a desired target perplexity $\mathcal{P}_{\text{target}}$. We use rejection sampling to ensure that the perplexity of the generated canaries falls within the range $[0.9\,\mathcal{P}_{\text{target}}, 1.1\,\mathcal{P}_{\text{target}}]$. We ensure the length is consistent across canaries, as this impacts memorization (Carlini et al., 2023; Kandpal et al., 2022). By adjusting the length of the in-distribution prefix, we can guide the generation of either entirely in-distribution or out-of-distribution canaries.

We insert each canary $n_{\text{rep}}$ times in the training dataset of target and reference models. When a canary is selected as a *member*, the canary is repeated $n_{\text{rep}}$ times in the training dataset, while canaries selected as *non-members* are excluded from the training dataset. As in prior work (Carlini et al., 2023; Kandpal et al., 2022; Meeus et al., 2024c), we opt for $n_{\text{rep}} > 1$ to increase memorization, thus facilitating privacy auditing and the observation of the effect of different factors on the performance of MIAs during ablation studies.

## 4 EXPERIMENTAL SETUP

**Datasets**  We consider two datasets that have been widely used to study text classification: (i) the Stanford Sentiment Treebank (**SST-2**) (Socher et al., 2013), which consists of excerpts from written movie reviews with a binary sentiment label; and (ii) the **AG News** dataset (Zhang et al., 2015), which consists of news articles labelled by category (World, Sport, Business, Sci/Tech). In all experiments, we remove examples with less than 5 words, bringing the total number of examples to $43\,296$ for SST-2 and $120\,000$ for AG News.

**Synthetic data generation**  We fine-tune the pre-trained Mistral-7B model (Jiang et al., 2023) using low-rank adaptation (LoRa) (Hu et al., 2022). We use a custom prompt template $\mathsf{p}(\cdot)$ for each dataset (see Appendix C). We sample synthetic data from the fine-tuned model $\theta$ conditioned on prompts $\mathsf{p}(\widetilde{\ell_i})$, following the same distribution of labels in the synthetic dataset $\widetilde{D}$ as in the original dataset $D$, i.e. $\ell_i = \widetilde{\ell}_i$ for $i = 1, ..., \widetilde{N}$. To generate synthetic sequences, we sequentially sample completions using a softmax temperature of $1.0$ and top-$p$ (aka nucleus) sampling with $p = 0.95$, i.e. we sample from a vocabulary restricted to the smallest possible set of tokens whose total probability exceeds $0.95$. We further ensure that the synthetic data we generate bears high utility, and is thus realistic. For this, we consider the downstream classification tasks for which the original datasets have been designed. We fine-tune RoBERTa-base (Liu et al., 2019) on $D$ and $\widetilde{D}$ and compare the performance of the resulting classifiers on held-out evaluation datasets. Further details and results are provided in Appendix E, for synthetic data generated with and without canaries.

| | Canary injection | | ROC AUC | | | |
|---|---|---|---|---|---|---|
| Dataset | Source | Label | Model | Synthetic (2-gram) | Synthetic $(\text{SIM}_{\text{Jac}})$ | Synthetic $(\text{SIM}_{\text{emb}})$ |
| SST-2 | In-distribution[1] | | 0.911 | 0.711 | 0.602 | 0.586 |
| | Synthetic | Natural | 0.999 | 0.616 | 0.547 | 0.530 |
| | | Artificial | 0.999 | 0.661 | 0.552 | 0.539 |
| AG News | In-distribution | | 0.993 | 0.620 | 0.590 | 0.565 |
| | Synthetic | Natural | 0.996 | 0.644 | 0.552 | 0.506 |
| | | Artificial | 0.999 | 0.660 | 0.560 | 0.525 |

Table 1: ROC AUC across training datasets, canary injection mechanisms and MIA methodologies. We give the ROC curves and TPR at low FPR scores in Appendix F, further ablations in Appendix G, and elaborate on the disparate vulnerability of high perplexity canaries in model- and data-based attacks in Appendix H.

**Canary injection**   We generate canaries $\hat{x} = (\hat{s}, \hat{\ell})$ as described in Sec. 3.2. Unless stated otherwise, we consider 50-word canaries. Synthetic canaries are generated using Mistral-7B (Jiang et al., 2023) as $\theta_0$. We consider two ways of constructing a canary label: (i) randomly sampling label $\hat{\ell}$ from the distribution of labels in the dataset, ensuring that the class distribution among canaries matches that of $D$ (*Natural*); (ii) extending the set of labels with a new artificial label ($\hat{\ell} =$"canary") only used for canaries (*Artificial*).

**Membership inference**   Throughout our experiments, we compute the $\beta_\theta(\hat{x})$ membership scores as described in Sec. 3.1. For one target model $\theta$, we consider 1000 canaries $\hat{x}$, of which on average half are included in the training dataset $n_{\text{rep}}$ times (members), while the remaining half are excluded (non-members). We then use the computed RMIA scores and the ground truth for membership to construct ROC curves for attacks from which we compute AUC and true positive rate (TPR) at low false positive rate (FPR) as measures of MIA performance. Across our experiments, we use $M = 4$ reference models $\theta'$, each trained on a dataset $D_{\theta'}$ consisting of the dataset $D$ used to train the target model $\theta$ with canaries inserted. Note that although practical attacks rarely have this amount of information, this is allowed by the threat model of Algorithm 1 and perfectly valid as a worst-case auditing methodology. We ensure that each canary is a member in half (i.e. 2) of the reference models and a non-member in the other half. For the attacks based on synthetic data, we use $n = 2$ for computing scores using an $n$-gram model and $k = 25$ for computing scores based on cosine similarity. In this latter case, we use Sentence-BERT (Reimers & Gurevych, 2019) (`paraphrase-MiniLM-L6-v2` from `sentence-transformers`) as the embedding model.

## 5   RESULTS

### 5.1   BASELINE EVALUATION WITH STANDARD CANARIES

We begin by assessing the vulnerability of synthetic text using standard canaries. Specifically, we utilize both in-distribution canaries and synthetically generated canaries with a target perplexity $\mathcal{P}_{\text{target}} = 250$, no in-distribution prefix ($F = 0$), $n_{\text{rep}} = 12$ and *natural* or *artificial* labels, as described in Section 4. Table 1 summarizes the ROC AUC for model- and data-based attacks.

First, we find that MIAs relying solely on the generated synthetic data achieve a ROC AUC score significantly higher than a random guess (i.e. AUC = 0.5), reaching up to 0.71 for SST-2 and 0.66 for AG News. This shows that synthetic text can leak information about the real data used to generate it.

---

[1]Constrained by in-distribution data, we here consider canaries of exactly 30 words (instead of 50 anywhere else).

Next, we observe that the data-based attack that uses an $n$-gram model trained on synthetic data to compute membership scores outperforms the two attacks that use instead similarity metrics: Jaccard distance between a canary and synthetic strings ($\mathrm{SIM_{Jac}}$) or cosine distance between their embeddings ($\mathrm{SIM_{emb}}$). This suggests that critical information for inferring membership lies in subtle changes in the co-occurrence of $n$-grams in synthetic data rather than in the generation of many sequences with lexical or semantic similarity.

We also compare attack performance across different canary types under data-based attacks $\mathcal{A}^{\widetilde{D}}$. The ROC AUC remains consistently higher than a random guess across all canaries. For SST-2, the highest AUC score of 0.71 is achieved when using in-distribution canaries. In contrast, for AG News, the highest AUC score of 0.66 is achieved for synthetic canaries with an artificial label not occurring in the dataset.

As another baseline, we test RMIA on the target model trained on $D$, under the assumption that the attacker has access to the model logits ($\mathcal{A}^{\theta}$). This attack achieves near-perfect performance across all setups, highlighting the fact that there is an inherent gap between the performance of model- and data-based attacks. A plausible explanation is that, while a fine-tuned model memorizes standard canaries well, the information necessary to infer their membership is partially transmitted to synthetic text generated using it.

To investigate the gap between the two attacks in more detail, we vary the number of canary repetitions $n_{\mathrm{rep}}$ to amplify the power of the data-based attack until its performance matches that of a model-based attack. Fig. 1(a) illustrates these results as a set of ROC curves. We quantify this discrepancy by noting that the MIA performance for $\mathcal{A}^{\widetilde{D}}$ at $n_{\mathrm{rep}} = 16$ is comparable to $\mathcal{A}^{\theta}$ at $n_{\mathrm{rep}} = 2$ and for low FPR at $n_{\mathrm{rep}} = 1$. We find similar results in Fig. 1(d) for AG News. The MIA performance for $\mathcal{A}^{\widetilde{D}}$ at $n_{\mathrm{rep}} = 16$ falls between the performance of $\mathcal{A}^{\theta}$ at $n_{\mathrm{rep}} = 1$ and $n_{\mathrm{rep}} = 2$. Under these experimental conditions, canaries would need to be repeated 8 to $16\times$ to reach the same vulnerability in data-based attacks compared to model-based attacks.

## 5.2 Designing specialized canaries for enhanced privacy auditing

To effectively audit privacy risks in a worst-case scenario, we explore the design of specialized canaries that are both memorized by the model and influential in the synthetic data.

First, we generate specialized canaries by controlling their target perplexity $\mathcal{P}_{\mathrm{target}}$. We evaluate MIAs for both threat models across a range of perplexities for canaries with natural labels, using $n_{\mathrm{rep}} = 4$ for the model-based attack $\mathcal{A}^{\theta}$ and $n_{\mathrm{rep}} = 16$ for the data-based attack $\mathcal{A}^{\widetilde{D}}$. We explore a wide range of perplexities, finding $1 \times 10^5$ to align with random token sequences. Figure 2 shows the ROC AUC score versus canary perplexity. For the model-based attack $\mathcal{A}^{\theta}$, the AUC monotonically increases with canary perplexity, reaffirming that outlier data records with higher perplexity are more vulnerable to MIAs (Feldman & Zhang, 2020; Carlini et al., 2022a; Meeus et al., 2024c). Conversely, for the data-based attack $\mathcal{A}^{\widetilde{D}}$, the AUC initially increases with perplexity but starts to decline beyond a certain threshold, eventually approaching a random guess (AUC of 0.5). To further illustrate this, we present the complete ROC curve in Figures 1(b) and (e) for SST-2 and AG News, respectively. We vary the canary perplexity $\mathcal{P}_{\mathrm{target}}$ while keeping other parameters constant. As $\mathcal{P}_{\mathrm{target}}$ increases, the model-based attack improves across the entire FPR range, while the data-based attack weakens, approaching a random guess at high perplexities. This suggests that identifying susceptible canaries is straightforward for model-based privacy audits, but assessing the privacy risk of synthetic data requires a careful balance between canary memorization and its influence on synthetic data.

We now examine whether canaries can be crafted to enhance both memorization and influence on the synthetic data, making them suitable to audit the privacy risks of releasing synthetic data. In Sec. 3.2, we introduced a method that exploits the greedy nature of LLM decoding to design more vulnerable canaries. We craft a canary with a low-perplexity in-distribution prefix to optimize its impact on the synthetic dataset, followed by a high-perplexity suffix to enhance memorization. We generate this suffix sampling from the pre-trained LLM $\theta_0$ with high temperature. Figures 1(c) and (f) illustrate the results for SST-2 and AG News, respectively. We

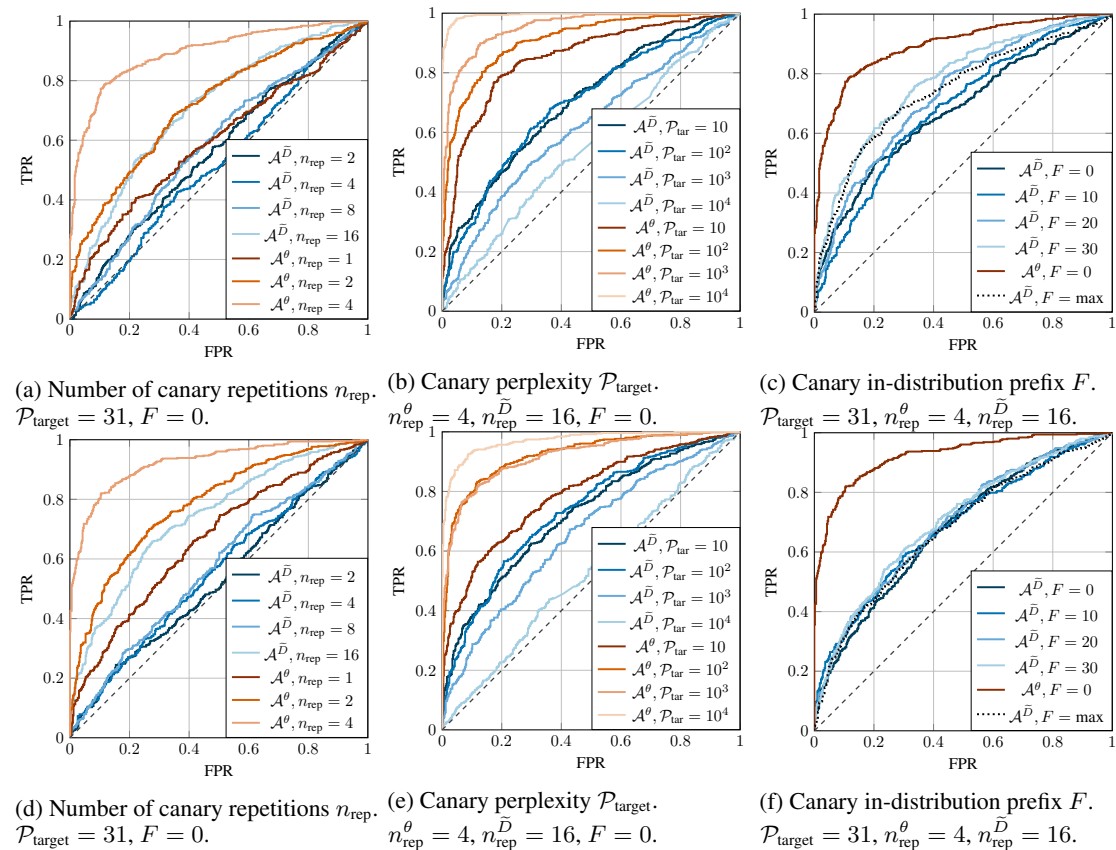

(a) Number of canary repetitions $n_{\text{rep}}$. $\mathcal{P}_{\text{target}} = 31$, $F = 0$.

(b) Canary perplexity $\mathcal{P}_{\text{target}}$. $n_{\text{rep}}^\theta = 4$, $n_{\text{rep}}^{\widetilde{D}} = 16$, $F = 0$.

(c) Canary in-distribution prefix $F$. $\mathcal{P}_{\text{target}} = 31$, $n_{\text{rep}}^\theta = 4$, $n_{\text{rep}}^{\widetilde{D}} = 16$.

(d) Number of canary repetitions $n_{\text{rep}}$. $\mathcal{P}_{\text{target}} = 31$, $F = 0$.

(e) Canary perplexity $\mathcal{P}_{\text{target}}$. $n_{\text{rep}}^\theta = 4$, $n_{\text{rep}}^{\widetilde{D}} = 16$, $F = 0$.

(f) Canary in-distribution prefix $F$. $\mathcal{P}_{\text{target}} = 31$, $n_{\text{rep}}^\theta = 4$, $n_{\text{rep}}^{\widetilde{D}} = 16$.

Figure 1: ROC curves of MIAs on synthetic data $\mathcal{A}^{\widetilde{D}}$ compared to model-based MIAs $\mathcal{A}^\theta$ on SST-2 ((a)–(c)) and AG News ((d)–(f)). We ablate over the number of canary insertions $n_{\text{rep}}$ in (a), (d), the target perplexity $\mathcal{P}_{\text{target}}$ of the inserted canaries in (b), (e) and the length $F$ of the in-distribution prefix in the canary in (c), (f).

set the overall canary perplexity $\mathcal{P}_{\text{target}} = 31$ and vary the prefix length $F$. As a reference, we also plot the results for in-distribution canaries labelled by $F = \text{max}$. We observe that combining an in-distribution prefix ($F > 0$) with a high-perplexity suffix ($F < \text{max}$) enhances attack effectiveness. This effect is especially notable for SST-2. For AG News, the improvement gained from adding an in-distribution prefix is less pronounced. This suggests that although the model's memorization of the canary stays consistent (as the overall perplexity remains unchanged), the canary's impact on the synthetic data becomes more prominent with longer in-distribution prefixes. We hypothesize that familiar low-perplexity prefixes serve as starting points for text generation, enhancing the likelihood that traces of the canary appear in the synthetic data.

## 6 RELATED WORK

**MIAs against ML models** Since the seminal work of Shokri et al. (2017), MIAs have been used to study memorization and privacy risks. Model-based MIAs have been studied under varying threat models, including adversaries with white-box access to model weights (Sablayrolles et al., 2019; Nasr et al., 2019; Leino & Fredrikson, 2020; Cretu et al., 2024), access to output probabilities (Shokri et al., 2017; Carlini et al., 2022a)

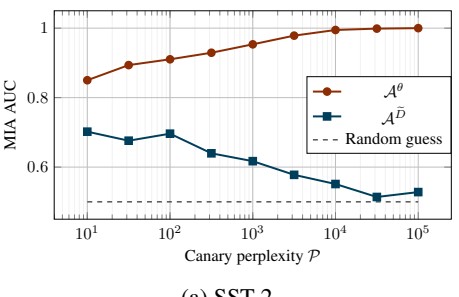 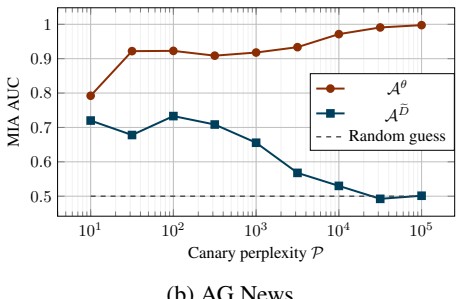

(a) SST-2  (b) AG News

Figure 2: ROC AUC score for synthetic canaries with varying perplexity (natural label). We present results for a model-based MIA $\mathcal{A}^{\theta}$ using output logits and a data-based attack $\mathcal{A}^{\widetilde{D}}$ using a 2-gram model. While the model-based attack improves as the perplexity increases, the inverse happens for the data-based attack.

or just labels (Choquette-Choo et al., 2021). The most powerful MIAs leverage a large number of reference models (Ye et al., 2022; Carlini et al., 2022a; Sablayrolles et al., 2019; Watson et al., 2021). Zarifzadeh et al. (2024) proposed RMIA, which achieves high performance using only a few.

**Attacks against language models**   Song & Shmatikov (2019) study the benign use of MIAs to audit the use of an individual's data during training. Carlini et al. (2021) investigate training data reconstruction attacks against LLMs. Kandpal et al. (2022) and Carlini et al. (2023) both study the effect of de-duplicating training data in reconstruction attacks by sampling a large corpus of synthetic text and running model-based attacks to identify likely members. Shi et al. (2024) and Meeus et al. (2024b) use attacks to identify pre-training data. Various membership inference scores have been proposed, such as the loss of target records (Yeom et al., 2018), lowest predicted token probabilities (Shi et al., 2024), changes in the model's probability for neighboring samples (Mattern et al., 2023), or perturbations to model weights (Li et al., 2023).

**MIAs against synthetic data in other scenarios**   Hayes et al. (2019) train a Generative Adversarial Network (GAN) on synthetic images generated by a target GAN and use the resulting discriminator to infer membership. Hilprecht et al. (2019) explore MIAs using synthetic images closest to a target record. Chen et al. (2020) study attack calibration techniques against GANs for images and location data. Privacy risks of synthetic tabular data have been widely studied, using MIAs based on similarity metrics and shadow models (Yale et al., 2019; Hyeong et al., 2022; Zhang et al., 2022). Stadler et al. (2022) compute high-level statistics, Houssiau et al. (2022) compute similarities between the target record and synthetic data, and Meeus et al. (2024a) propose a trainable feature extractor. Unlike these, we evaluate MIAs on text generated using fine-tuned LLMs. This introduces unique challenges and opportunities, both in computing membership scores and identifying worst-case canaries, making our approach distinct from prior work.

**Vulnerable records in MIAs**   Prior work established that some records (*outliers*) have a disparate effect on a trained model compared to others (Feldman & Zhang, 2020), making them more vulnerable to MIAs (Carlini et al., 2022a;b). Hence, specifically crafted canaries have been proposed to study memorization and for privacy auditing of language models, ranging from a sequence of random digits (Carlini et al., 2019; Stock et al., 2022) or random tokens (Wei et al., 2024) to synthetically generated sequences (Meeus et al., 2024c). In the case of synthetic tabular data, Stadler et al. (2022) find that statistical outliers have increased privacy leakage, while Meeus et al. (2024a) propose measuring the distance to the closest records to infer membership.

**Decoding method**   We use fixed prompt templates and top-$p$ sampling to assess the privacy of synthetic text in a realistic regime rather than allowing the attacker to pick a decoding method adversarially. Research on data reconstruction attacks study how decoding methods like beam search (Zanella-Béguelin et al., 2020; Carlini et al., 2023), top-$k$ sampling (Kandpal et al., 2022), or decaying temperature (Carlini et al., 2021) impact how often LLMs replicate information from their training data.

## 7 REPRODUCIBILITY STATEMENT

Both datasets used in this paper are publicly available (Socher et al., 2013; Zhang et al., 2015), and so is the pre-trained model (Jiang et al., 2023) we used. We fine-tune the pre-trained model for 1 epoch using LoRA with $r = 4$, including all target modules (10.7M parameters in total). We use an effective batch size of 128 and learning rate $\eta = 2 \times 10^{-5}$ (for more details see Appendix J). All our experiments have been conducted on a cluster of nodes with 8 V100 NVIDIA GPUs with a floating point precision of 16 (`fp16`). We built our experiments on two open-source packages: (i) `privacy-estimates` which provides a distributed implementation of the RMIA attack and (ii) `dp-transformers` which provides the implementation of the synthetic data generator. All of our code is attached in the supplemented materials. In addition, we will release the code necessary to reproduce the results presented in this paper on GitHub upon publication.

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

## A    PSEUDO-CODE FOR MIAS BASED ON SYNTHETIC DATA

We here provide the pseudo-code for computing membership signals for both MIA methodologies based on synthetic data (Sec. 3.1.2), see Algorithm 2 for the $n$-gram method and Algorithm 3 for the method using similarity metrics.

---

**Algorithm 2** Compute membership signal using $n$-gram model

---

1: **Parameter**: $n$-gram model order $n$
2: **Input**: Synthetic dataset $\widetilde{D} = \{\widetilde{x}_i = (\widetilde{s}_i, \widetilde{\ell}_i)\}_{i=1}^{\widetilde{N}}$, Target canary $\hat{x} = (\hat{s}, \hat{\ell})$
3: **Output**: Membership signal $\alpha$
4: $C(\vec{w}) \leftarrow 0$ for all $(n-1)$- and $n$-grams $\vec{w}$
5: **for** $i = 1$ to $\widetilde{N}$ **do**
6: $\quad w_1, \ldots, w_{k(i)} \leftarrow \widetilde{s}_i$
7: $\quad$ **for** each $n$-gram $(w_{j-(n-1)}, \ldots, w_j)$ in $\widetilde{s}_i$ **do**
8: $\quad\quad C(w_{j-(n-1)}, \ldots, w_j) \mathrel{+}= 1$
9: $\quad\quad C(w_{j-(n-1)}, \ldots, w_{j-1}) \mathrel{+}= 1$
10: $V \leftarrow |\{w \mid \exists i.w \in \widetilde{s}_i\}|$
11: The $n$-gram model is factored into conditional probabilities: $\qquad\qquad\triangleright$ *Final $n$-gram model*

$$P_{n\text{-gram}}(w_j \mid w_{j-(n-1)}, \ldots, w_{j-1}) = \frac{C(w_{j-(n-1)}, \ldots, w_j) + 1}{C(w_{j-(n-1)}, \ldots, w_{j-1}) + V}$$

12: $w_1, \ldots, w_k \leftarrow \hat{s}$ $\qquad\qquad\qquad\qquad\qquad\triangleright$ *Compute probability of canary text $\hat{s}$*
13: $\alpha \leftarrow \prod_{j=2}^{k} P_{n\text{-gram}}(w_j \mid w_{j-(n-1)}, \ldots, w_{j-1})$
14: **return** $\alpha$

---

---

**Algorithm 3** Compute membership signal using similarity metric

---

1: **Parameter**: Similarity metric $\text{SIM}(\cdot, \cdot)$, cutoff parameter $k$
2: **Input**: Synthetic dataset $\widetilde{D} = \{\widetilde{x}_i = (\widetilde{s}_i, \widetilde{\ell}_i)\}_{i=1}^{\widetilde{N}}$, Target canary $\hat{x} = (\hat{s}, \hat{\ell})$
3: **Output**: Membership signal $\alpha$
4: **for** $i = 1$ to $\widetilde{N}$ **do** $\qquad\qquad\qquad\qquad\triangleright$ *Compute similarity of each synthetic example*
5: $\quad \sigma_i \leftarrow \text{SIM}(\hat{s}, \widetilde{s}_i)$
6: Sort similarities $\sigma_i$ for $i = 1, \ldots, \widetilde{N}$ in descending order
7: Let $\sigma_{i(1)}, \ldots, \sigma_{i(k)}$ be the top-$k$ similarities
8: $\alpha \leftarrow \frac{1}{k} \sum_{j=1}^{k} \sigma_{i(j)}$ $\qquad\qquad\qquad\triangleright$ *Compute mean similarity of the top-$k$ examples*
9: **return** $\alpha$

---

## B    COMPUTATION OF RMIA SCORES

We here provide more details on how we adapt RMIA, as originally proposed by Zarifzadeh et al. (2024), to our setup (see Sec. 3.1.3). In RMIA, the pairwise likelihood ratio is defined as:

$$LR_\theta(x, z) = \left(\frac{P(x \mid \theta)}{P(x)}\right) \left(\frac{P(z \mid \theta)}{P(z)}\right)^{-1} . \tag{3}$$

where $\theta$ represents the target model, $x$ the target record, and $z$ the reference population. In this work, we only consider one target model $\theta$ and many target records $x$. As we are only interested in the relative value of the likelihood ratio across target records, we can eliminate the dependency on the reference population $z$,

$$LR_\theta(x, z) = LR_\theta(x) = \frac{P(x \mid \theta)}{P(x)} . \tag{4}$$

As suggested by Zarifzadeh et al. (2024), we compute $P(x)$ as the empirical mean of $P(x \mid \theta')$ across reference models $\{\theta'_i\}_{i=1}^M$,

$$P(x) = \frac{1}{M} \sum_{i=1}^{M} P(x \mid \theta'_i) . \tag{5}$$

To compute RMIA scores, we replace the probabilities in (4) by membership signals on target and reference models:

$$\beta_\theta(x) = \frac{\alpha_\theta(x)}{\frac{1}{M} \sum_{i=1}^{M} \alpha_{\theta'_i}(x)} . \tag{6}$$

Note that when we compute $\alpha_\theta(x)$ as a product of conditional probabilities (e.g. when using the target model probability in the model-based attack or the $n$-gram probability in the data-based attack), we truly use a probability for $\alpha_\theta(x)$. However, in the case of the data-based attack using similarity metrics, we use the mean similarity to the $k$ closest synthetic sequences—which does not correspond to a true probability. In this case, we normalize similarities to fall in the range $[0, 1]$ and use $\alpha_\theta(x)$ as an empirical proxy for the probability $P(x \mid \theta)$.

In practice, $P(x \mid \theta)$ can be an extremely small value, particularly when calculated as a product of token-level conditional probabilities, which can lead to underflow errors. To mitigate this, we perform arithmetic operations on log-probabilities whenever possible. However, in the context of equation (6), where the denominator involves averaging probabilities, we employ quad precision floating-point arithmetic. This method is sufficiently precise to handle probabilities for sequences of up to 50 words, which is the maximum we consider in our experiments.

## C    PROMPTS USED TO GENERATE SYNTHETIC DATA

Table 2 summarizes the prompt templates $\mathsf{p}(\ell)$ used to generate synthetic data for both datasets (see Sec. 4).

| Dataset | Template $\mathsf{p}(\ell)$ | Labels $\ell$ |
|---|---|---|
| SST-2 | "This is a sentence with a $\ell$ sentiment: " | {positive, negative} |
| AG News | "This is a news article about $\ell$: " | {World, Sport, Business, Sci/Tech} |

Table 2: Prompt templates used to fine-tune models and generate synthetic data.

## D    DETAILED ASSUMPTIONS MADE FOR THE ADVERSARY

We clarify the capabilities of adversaries in model- and data-based attacks according to the threat model specified in Section 2. We note:

1. A model-based attack is strictly more powerful than a data-based attack. This is because with access to the fine-tuned model $\theta$ and the prompt template $\mathsf{p}(\cdot)$, a model-based attack can synthesize $\widetilde{\mathcal{D}}$ for any set of synthetic labels and perfectly simulate the membership inference experiment for a data-based attack.

2. In both threat models, the adversary can train reference models $\{\theta'_i\}_{i=1}^M$. This assumes access to the private dataset $D$, and the training procedure of target model $\theta$, including hyperparameters. This is made clear in line 3 in Algorithm 1.

3. In our experiments, we consider model-based attacks that use the prompt template $\mathsf{p}(\cdot)$ to compute the model loss for target records, as specified in Sec. 3.1.1. Our data-based attacks use the prompt template $\mathsf{p}(\cdot)$ to generate synthetic data $\widetilde{D}$ from reference models.

4. Only the model-based attack has query-access to the target model $\theta$. The attacks used in our experiments use $\theta$ to compute token-level predicted logits for input sequences and do not use white-box features, although this is not excluded by the threat model.

5. Only the data-based attack generates synthetic data from reference models, so only this threat model leverages the sampling procedure $\mathsf{sample}(\cdot)$.

Table 3 summarizes the adversary capabilities used in the attacks in our experiments.

| Assumptions | Model-based MIA | Data-based MIA |
|---|---|---|
| Knowledge of the private dataset $D$ used to fine-tune the target model $\theta$ (apart from knowledge of canaries). | ✓ | ✓ |
| Knowledge of the training procedure of target model $\theta$. | ✓ | ✓ |
| Knowledge of the prompt template $\mathsf{p}(\ell_i)$ used to generate the synthetic data. | ✓ | ✓ |
| Query-access to target model $\theta$, returning predicted logits. | ✓ | – |
| Access to synthetic data $\widetilde{D}$ generated by target model $\theta$. | – | ✓ |
| Knowledge of the decoding strategy employed to sample synthetic data $\widetilde{D}$ (e.g., temperature, top-$k$). | – | ✓ |

Table 3: Adversary capabilities effectively used by attacks in our experiments.

# E  SYNTHETIC DATA UTILITY

To ensure we audit the privacy of synthetic text data in a realistic setup, the synthetic data needs to bear high utility. We measure the synthetic data utility by comparing the downstream classification performance of RoBERTa-base (Liu et al., 2019) when fine-tuned exclusively on real or synthetic data. We fine-tune models for binary (SST-2) and multi-class classification (AG News) for 1 epoch on the same number of real or synthetic data records using a batch size of 16 and learning rate $\eta = 1 \times 10^{-5}$. We report the macro-averaged AUC score and accuracy on a held-out test dataset of real records.

Table 4 summarizes the results for synthetic data generated based on original data which does not contain any canaries. While we do see a slight drop in downstream performance when considering synthetic data instead of the original data, AUC and accuracy remain high for both tasks.

We further measure the synthetic data utility when the original data contains standard canaries (see Sec. 5.1). Specifically, we consider synthetic data generated from a target model trained on data containing 500 canaries

| Dataset | Fine-tuning data | Classification | |
|---|---|---|---|
| | | AUC | Accuracy |
| SST-2 | Real | 0.984 | 92.3 % |
| | Synthetic | 0.968 | 91.5 % |
| AG News | Real | 0.992 | 94.4 % |
| | Synthetic | 0.978 | 90.0 % |

Table 4: Utility of synthetic data generated from real data *without* canaries. We compare the performance of text classifiers trained on real or synthetic data—both evaluated on real, held-out test data.

repeated $n_{\text{rep}} = 12$ times, so 6000 data records. When inserting canaries with an artificial label, we remove all synthetic data associated with labels not present originally when fine-tuning the RoBERTa-base model.

| Dataset | Canary injection | | Classification | |
|---|---|---|---|---|
| | Source | Label | AUC | Accuracy |
| SST-2 | In-distribution | | 0.972 | 91.6 % |
| | Synthetic | Natural | 0.959 | 89.3 % |
| | | Artificial | 0.962 | 89.9 % |
| AG News | In-distribution | | 0.978 | 89.8 % |
| | Synthetic | Natural | 0.977 | 88.6 % |
| | | Artificial | 0.980 | 90.1 % |

Table 5: Utility of synthetic data generated from real data *with* canaries ($n_{\text{rep}} = 12$). We compare the performance of text classifiers trained on real or synthetic data—both evaluated on real, held-out test data.

Table 5 summarizes the results. Across all canary injection methods, we find limited impact of canaries on the downstream utility of synthetic data. While the difference is minor, the natural canary labels lead to the largest utility degradation. This makes sense, as the high perplexity synthetic sequences likely distort the distribution of synthetic text associated with a certain real label. In contrast, in-distribution canaries can be seen as up-sampling certain real data points during fine-tuning, while canaries with artificial labels merely reduce the capacity of the model to learn from real data and do not interfere with this process as much as canaries with natural labels do.

## F    ADDITIONAL RESULTS FOR MIAS USING STANDARD CANARIES

In line with the literature on MIAs against machine learning models (Carlini et al., 2022a), we also evaluate MIAs by their true positive rate (FPR) at low false positive rates (FPR). Tables 6 and 7 summarize the MIA TPR at FPR=0.01 and FPR=0.1, respectively. We also provide the ROC curves for the MIAs for both datasets (with canary labels randomly sampled from the distribution of labels in real data) in Figure 3.

## G    ABLATIONS FOR MIAS ON SYNTHETIC DATA

**Synthetic multiple**    Thus far, we have exclusively considered that the number of generated synthetic records equals the number of records in the real data, i.e., $N = \widetilde{N}$. We now consider the case when more synthetic data is made available to a data-based adversary ($\widetilde{\mathcal{A}}$). Specifically, we denote the *synthetic multiple* $m = \widetilde{N}/N$

| Dataset | Canary injection | | TPR@FPR=0.01 | | | |
| | Source | Label | Model | Synthetic (2-gram) | Synthetic ($\text{SIM}_{\text{Jac}}$) | Synthetic ($\text{SIM}_{\text{emb}}$) |
|---|---|---|---|---|---|---|
| SST-2 | In-distribution | | 0.148 | 0.081 | 0.029 | 0.020 |
| | Synthetic | Natural | 0.972 | 0.032 | 0.018 | 0.024 |
| | | Artificial | 0.968 | 0.049 | 0.000 | 0.030 |
| AG News | In-distribution | | 0.941 | 0.063 | 0.032 | 0.016 |
| | Synthetic | Natural | 0.955 | 0.030 | 0.006 | 0.016 |
| | | Artificial | 0.990 | 0.071 | 0.041 | 0.022 |

Table 6: True positive rate (TPR) at a false positive rate (FPR) of 0.01 for experiments using standard canaries (Sec. 5.1) across training datasets, canary injection mechanisms and MIA methodologies. Canaries are synthetically generated with target perplexity $\mathcal{P}_{\text{target}} = 250$ and inserted $n_{\text{rep}} = 12$ times.

| Dataset | Canary injection | | TPR@FPR=0.1 | | | |
| | Source | Label | Model | Synthetic (2-gram) | Synthetic ($\text{SIM}_{\text{Jac}}$) | Synthetic ($\text{SIM}_{\text{emb}}$) |
|---|---|---|---|---|---|---|
| SST-2 | In-distribution | | 0.795 | 0.335 | 0.207 | 0.203 |
| | Synthetic | Natural | 0.996 | 0.209 | 0.114 | 0.128 |
| | | Artificial | 1.000 | 0.268 | 0.142 | 0.142 |
| AG News | In-distribution | | 0.982 | 0.200 | 0.158 | 0.168 |
| | Synthetic | Natural | 0.990 | 0.260 | 0.114 | 0.114 |
| | | Artificial | 0.996 | 0.298 | 0.152 | 0.164 |

Table 7: True positive rate (TPR) at a false positive rate (FPR) of 0.1 for experiments using standard canaries (Sec. 5.1) across training datasets, canary injection mechanisms and MIA methodologies. Canaries are synthetically generated with target perplexity $\mathcal{P}_{\text{target}} = 250$ and inserted $n_{\text{rep}} = 12$ times.

and evaluate how different MIAs perform for varying values of $m$. Figure 4 shows how the ROC AUC score varies as $m$ increases. As expected, the ROC AUC score for the attack that uses membership signals computed using a 2-gram model trained on synthetic data increases when more synthetic data is available. In contrast, attacks based on similarity metrics do not seem to benefit significantly from this additional data.

**Hyperparameters in model-based attacks** The model-based attacks that we presented in Sec. 3.1 have hyperparameters. The attack that uses $n$-gram models to compute membership signals is parameterized by the order $n$. Using a too small value for $n$ might not suffice to capture the information leaked from canaries into the synthetic data used to train the $n$-gram model. When using a too large order $n$, on the other hand, we would expect less overlap between $n$-grams present in the synthetic data and the canaries, lowering the membership signal.

Further, the similarity-based methods rely on the computation of the mean similarity of the closest $k$ synthetic records to the a canary. When $k$ is very small, e.g. $k = 1$, the method takes into account a single synthetic record, potentially missing on leakage of membership information from other close synthetic data records. When $k$ becomes too large, larger regions of the synthetic data in embedding space are taken into account, which might dilute the membership signal among the noise.

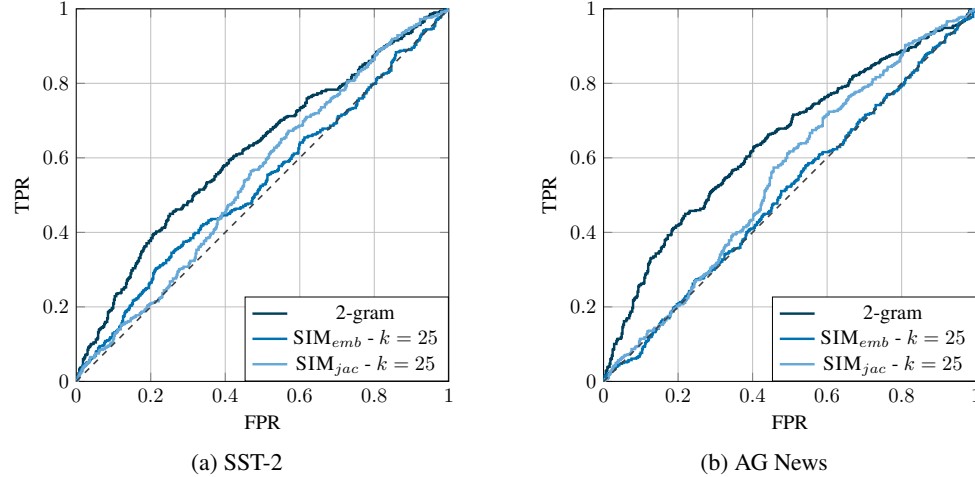

(a) SST-2

(b) AG News

Figure 3: MIA ROC curves across MIA methodologies for the SST-2 (left) and AG News (right) datasets. Canaries are synthetically generated with target perplexity of $\mathcal{P}_{\text{target}} = 250$ with a natural label and inserted $n_{\text{rep}} = 12$ times.

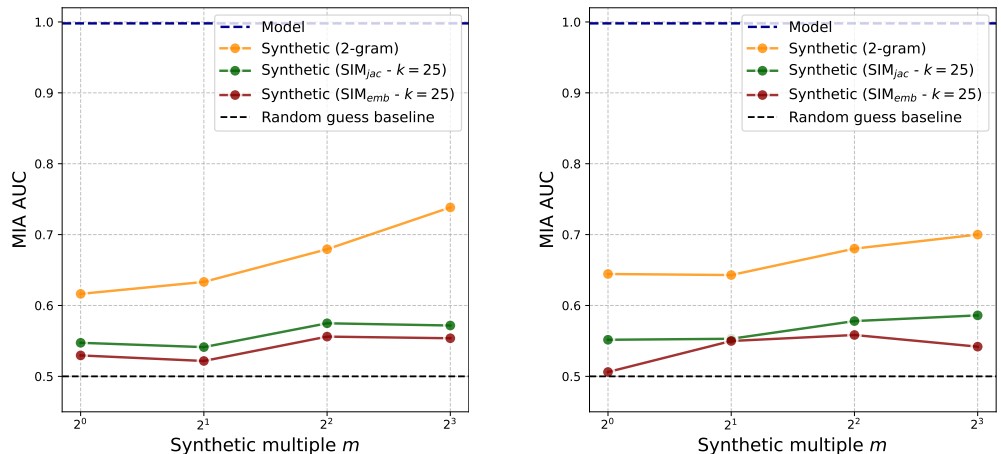

Figure 4: ROC AUC score for increasing value of the synthetic multiple $m$ across model-based attack methods for SST-2 (left) and AG News (right). Canaries are synthetically generated with target perplexity of $\mathcal{P}_{\text{target}} = 250$, with a natural label, and inserted $n_{\text{rep}} = 12$ times.

Table 8 reports the ROC AUC scores of model-based attacks for different values of the hyperparameters $n$ and $k$ when using standard canaries (Sec. 5.1).

## H    DISPARATE VULNERABILITY OF STANDARD CANARIES

We analyze the disparate vulnerability of standard canaries between the model-based attack and the data-based attack that uses a 2-gram model (as discussed in Sec 5.1). Figure 5 plots the RMIA scores for both attacks on the same set of canaries, which have either been included in the training dataset of the target model (*member*)

| Dataset | $n$-gram | | SIM$_{\text{Jac}}$ | | SIM$_{\text{emb}}$ | |
|---|---|---|---|---|---|---|
| | $n$ | AUC | $k$ | AUC | $k$ | AUC |
| SST-2 | 1 | 0.415 | 1 | 0.520 | 1 | 0.516 |
| | 2 | **0.616** | 5 | 0.535 | 5 | 0.516 |
| | 3 | 0.581 | 10 | 0.538 | 10 | 0.519 |
| | 4 | 0.530 | 25 | **0.547** | 25 | **0.530** |
| AG News | 1 | 0.603 | 1 | 0.522 | 1 | 0.503 |
| | 2 | **0.644** | 5 | 0.525 | 5 | 0.498 |
| | 3 | 0.567 | 10 | 0.537 | 10 | 0.503 |
| | 4 | 0.527 | 25 | **0.552** | 25 | **0.506** |

Table 8: Ablation over hyperparameters of model-based MIAs. We report ROC AUC scores across different values of the hyperparameters $n$ and $k$ (see Sec. 3.1). Canaries are synthetically generated with target perplexity $\mathcal{P}_{\text{target}} = 250$, with a natural label, and inserted $n_{\text{rep}} = 12$ times.

or not (*non-member*). Note that the RMIA scores are used to distinguish members from non-members, and that a larger value corresponds to the adversary being more confident in identifying a record as a member, i.e., to the record being more *vulnerable*.

First, we note that the scores across both threat models exhibit a statistically significant, positive correlation. We find a Pearson correlation coefficient between the RMIA scores (log) for both methods of $0.20$ ($p$-value of $2.4 \times 10^{-10}$) and $0.23$ ($p$-value of $1.9 \times 10^{-13}$) for SST-2 and AG News, respectively. This means that a record vulnerable to the model-based attack tends to be also vulnerable to the data-based attack, even though the attacks differ substantially.

Second, and more interestingly, some canaries have disparate vulnerability across MIA methods. Indeed, Figure 5 shows how certain data records which are not particularly vulnerable to the model-based attack are significantly more vulnerable to the data-based attack, and vice versa.

# I LOW FPR ROC RESULTS

Figure 6 shows log-log plots of the ROC curves in Figure 1 to better examine behavior of attacks at low FPR.

# J DETERMINING OPTIMAL HYPERPARAMETERS

We optimized hyperparameters for LoRA fine-tuning Mistral-7B on SST-2 by running a grid search over learning rate ($[1 \times 10^{-6}, 4 \times 10^{-6}, 2 \times 10^{-5}, 6 \times 10^{-5}, 3 \times 10^{-4}, 1 \times 10^{-3}]$) and batch size ($[64, 128, 256]$). We fine-tuned the models for 3 epochs and observed the validation loss plateaued after the first epoch. Based on these results, we selected a learning rate of $2 \times 10^{-5}$, effective batch size of 128, sequence length 128, LoRA $r = 4$ and fine-tuned the models for 1 epoch, as stated in Sec. 7. Figure 7 shows the validation cross-entropy loss for SST-2 over the grid we searched on and the train and validation loss curves for 3 epochs with the selected hyperparameters.

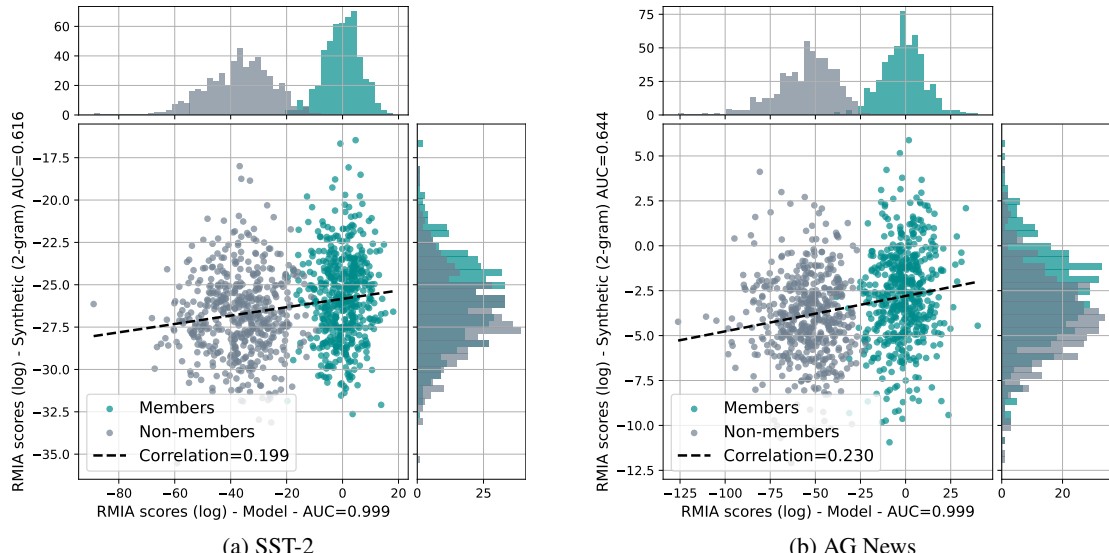

(a) SST-2                                                    (b) AG News

Figure 5: RMIA scores (log) for model- and data-based MIAs on the same set of canaries. Results for both datasets SST-2 and AG News. Canaries are synthetically generated with target perplexity of $\mathcal{P}_{\text{target}} = 250$ with a natural label, and inserted $n_{\text{rep}} = 12$ times.

## K    INTERPRETABILITY

### K.1    IDENTIFYING MEMORIZED SUB-SEQUENCES

We analyze what information from a canary leaks into the synthetic data that enables a data-based attack to infer its membership. For each canary $\hat{x} = (\hat{s}, \hat{\ell})$, we examine the synthetic data generated by a model trained on a dataset including (member) and excluding $\hat{x}$ (non-member). We leverage the $M = 4$ reference models $\theta'$ used to develop the attack for 1000 specialized canaries from Fig. 1(c). For each model $\theta'$, we count the number of $n$-grams in $\widetilde{s}$ that occur at least once in $\widetilde{D}'$ ($C_{\text{unique}}$). We also compute the median $C_{\text{med}}$ and average $C_{\text{avg}}$ counts of $n$-grams from $\hat{s}$ in $\widetilde{D}'$. Table 9 summarizes how these measures vary with $n$. As $n$ increases, the number of $n$-grams from the canary appearing in the synthetic data drops sharply, reaching $C_{\text{med}} = 0$ for $n = 4$ for models including and excluding a canary. This suggests that any verbatim reproduction of canary text in the generated synthetic data is of limited length. Further, we observe only slight differences in counts between members and non-members, indicating that the signal for inferring membership is likely in subtle shifts in the probability distribution of token co-occurrences within the synthetic data, as captured by the 2-gram model. We further analyze canaries with the highest and lowest RMIA scores below.

### K.2    INTERPRETABILITY OF RMIA SCORES

To further understand the membership signal for data-based attacks, we examine some examples in-depth.

Specifically, we consider the MIA for specialized canaries with $F = 30$, $\mathcal{P}_{\text{target}} = 31$ and $n_{\text{rep}} = 16$ for SST-2 from Figure 1(c). Recall that for this attack, we consider 1000 canaries, 500 of which are injected into the training dataset of one target model $\theta$. We also train 4 references models $\{\theta_i'\}_{i=1}^4$ where each of the 1000 canaries has been included in exactly half. We focus on the best performing MIA based on synthetic data, i.e.

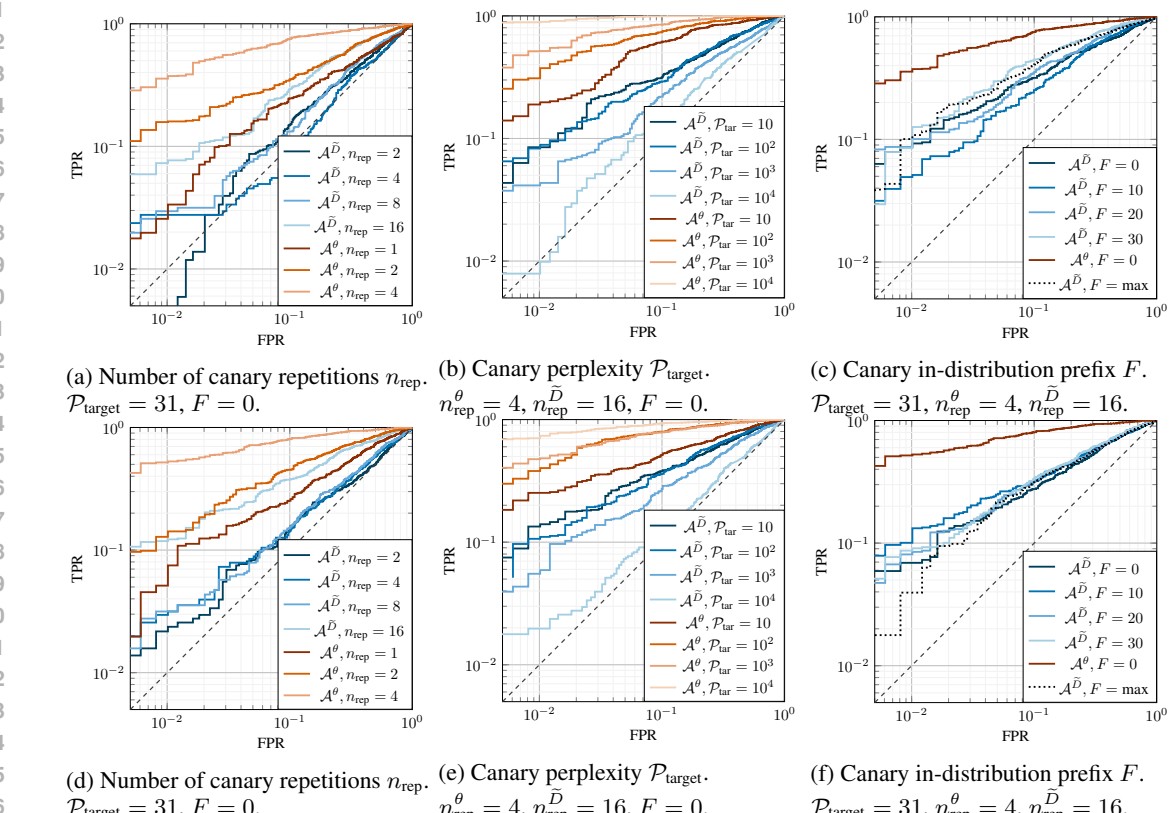

(a) Number of canary repetitions $n_{\text{rep}}$. $\mathcal{P}_{\text{target}} = 31$, $F = 0$.

(b) Canary perplexity $\mathcal{P}_{\text{target}}$. $n_{\text{rep}}^\theta = 4$, $n_{\text{rep}}^{\widetilde{D}} = 16$, $F = 0$.

(c) Canary in-distribution prefix $F$. $\mathcal{P}_{\text{target}} = 31$, $n_{\text{rep}}^\theta = 4$, $n_{\text{rep}}^{\widetilde{D}} = 16$.

(d) Number of canary repetitions $n_{\text{rep}}$. $\mathcal{P}_{\text{target}} = 31$, $F = 0$.

(e) Canary perplexity $\mathcal{P}_{\text{target}}$. $n_{\text{rep}}^\theta = 4$, $n_{\text{rep}}^{\widetilde{D}} = 16$, $F = 0$.

(f) Canary in-distribution prefix $F$. $\mathcal{P}_{\text{target}} = 31$, $n_{\text{rep}}^\theta = 4$, $n_{\text{rep}}^{\widetilde{D}} = 16$.

Figure 6: Log-log ROC curves of MIAs on synthetic data $\mathcal{A}^{\widetilde{D}}$ compared to model-based MIAs $\mathcal{A}^\theta$ on SST-2 ((a)–(c)) and AG News ((d)–(f)). We ablate over the number of canary insertions $n_{\text{rep}}$ in (a), (d), the target perplexity $\mathcal{P}_{\text{target}}$ of the inserted canaries in (b), (e) and the length $F$ of the in-distribution prefix in the canary in (c), (f).

the attack leveraging the probability of the target sequence computed using a 2-gram model trained on the synthetic data.

| | $C_{\text{unique}}$ | | $C_{\text{med}}$ | | $C_{\text{avg}}$ | |
|---|---|---|---|---|---|---|
| $n$ | Member | Non-member | Member | Non-member | Member | Non-member |
| 1 | $46.1 \pm 2.5$ | $45.2 \pm 2.8$ | $882.9 \pm 756.3$ | $884.2 \pm 771.8$ | $7391.0 \pm 1892.23$ | $7382.7 \pm 1887.1$ |
| 2 | $29.6 \pm 5.7$ | $28.1 \pm 5.7$ | $5.2 \pm 6.6$ | $4.2 \pm 6.3$ | $202.9 \pm 118.0$ | $199.6 \pm 116.6$ |
| 4 | $4.8 \pm 3.6$ | $3.9 \pm 3.2$ | $0.0 \pm 0.0$ | $0.0 \pm 0.0$ | $1.4 \pm 2.8$ | $1.2 \pm 2.6$ |
| 8 | $0.1 \pm 0.6$ | $0.0 \pm 0.3$ | $0.0 \pm 0.0$ | $0.0 \pm 0.0$ | $0.0 \pm 0.0$ | $0.0 \pm 0.0$ |

Table 9: Aggregate count statistics of $n$-grams in a canary $\hat{s}$ that also appear in the synthetic data $\widetilde{D}'$ generated using 4 reference models including and excluding $\hat{s}$. Number of $n$-grams in $\widetilde{s}$ that also appear in $\widetilde{D}'$ ($C_{\text{unique}}$), median ($C_{\text{med}}$) and average ($C_{\text{avg}}$) counts of $n$-grams from $\hat{s}$ in $\widetilde{D}'$. We report mean and std. deviation of these measures over all canaries ($F = 30$, $\mathcal{P}_{\text{target}} = 31$, $n_{\text{rep}} = 16$) for SST-2. Each canary $\hat{s}$ contains exactly 50 words and $\widetilde{D}'$ contains $706.7k \pm 72.8k$ words.

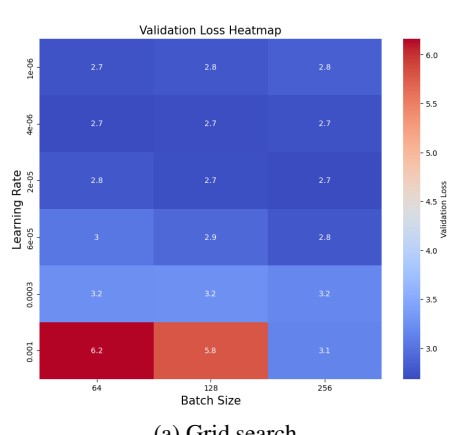

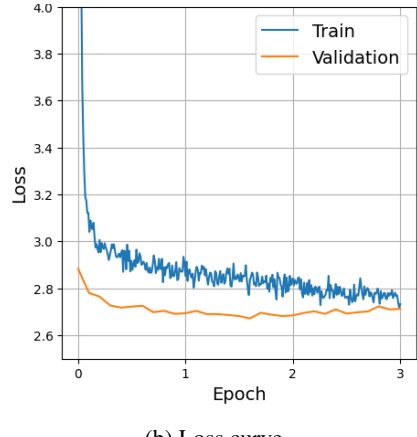

(a) Grid search                                           (b) Loss curve

Figure 7: (a) Validation cross-entropy loss of LoRA fine-tuning Mistral-7B on SST-2 varying the learning rate and effective batch size. (b) Training and validation loss for best hyperparameters over 3 epochs.

To understand what signal the MIA picks up to infer membership, we focus on the canary most confidently—and correctly—identified as member and the one most confidently—and correctly—identified as non-member. For this, we take the canaries for which the RMIA score computed using the target model and the reference models is the highest and the lowest, respectively.

Next, for each model (4 reference models, and 1 target model), we report for this canary $\hat{x}_i$:

1. Whether the canary has been included in, $\hat{x}_i \in D$ (IN), or excluded from, $\hat{x}_i \notin D$ (OUT), the training dataset of the model in question, and thus to generate the synthetic data $\widetilde{D} = \{\widetilde{x}_i = (\widetilde{s}_i, \widetilde{\ell}_i)\}_{i=1}^{\widetilde{N}}$.

2. The canary with the words that appear as a 2-gram in the synthetic data $\widetilde{D}$ emphasized in bold face. Note that if, for instance, this is a sequence of 3 words, e.g., *"like many western"*, this means that all 3 words appear in 2-grams in the synthetic data, e.g., *"like many"* and *"many western"*.

3. The maximum overlapping sub-string between the canary and any synthetically generated record $\widetilde{s}_i$. We define a sub-string as a sequence of characters, including white space, and also report its length as number of characters $L_{\text{overlap}}$.

4. The mean, negative cross-entropy loss of the canary computed using the 2-gram model trained on the synthetic data. Formally, for canary $\hat{s}_i = (w_1, w_2, \ldots, w_k)$: $-\frac{1}{k} \sum_{j=2}^{k} \log \left( P_{\text{2-gram}}(w_j, w_{j-1}) \right)$.

Tables 10 and 11 report this for the canary with the largest and lowest RMIA score, respectively.

First, we observe that not all the words in the canary appear as 2-grams in the synthetic dataset. This could be expected, as not all 2-grams are commonly used in general English (e.g. *"penetrating views"*). Notably, the number of common 2-grams does not significantly differ whether the canary is a member or not (IN or OUT).

In addition, we observe similar trends when considering the longest overlapping sub-string between the canary and the synthetic data. Across all models and canaries, this sub-string remains consistently short and shows little variation with membership labels. This suggests that the signal used to infer membership does not rely on the verbatim regurgitation of long sub-sequences.

Lastly, we investigate whether the reported 2-gram loss is consistent with the fact that these canaries correspond to the largest and lowest RMIA scores. Although the losses across models differ only slightly, the relative values align with the RMIA scores. Recall that RMIA scores are intuitively computed as the ratio of

the membership signal of the target model to the average membership signal across reference models. For the canary with the highest RMIA score, the 2-gram loss of the target model is lower than the average loss of the reference models, suggesting that the canary was seen by the target model. Conversely, for the canary with the lowest RMIA score, the 2-gram loss is higher than the average loss across reference models.

These results suggest that the information required to infer membership based on synthetic data does not lie in the explicit generation of canary sub-strings within the synthetic data. Instead, the signal seems more subtle, arising from slight shifts in the probability distribution of co-occurrences of words in the synthetic data.

| Model | IN or OUT | Canary (words present as part of 2-grams in $\widetilde{D}'$ in bold) | Max overlapping sub-string | 2-gram loss |
|---|---|---|---|---|
| $\theta'_1$ (ref) | IN | "**like many western action films , this thriller is too loud and thoroughly overbearing , but its heartfelt** concern about north korea **'s recent past and south korea** 's future, **its sophisticated sense of character and its** penetrating views **on many social and political** issues, **like the** exploitation **of single**" | « *social and political issues* » ; $L_{\text{overlap}} = 28$ | 17.96 |
| $\theta'_2$ (ref) | IN | "**like many western action films , this thriller is too loud and thoroughly** overbearing , **but its heartfelt concern about north korea 's recent past and south korea** 's future, its **sophisticated sense of character and its** penetrating views **on many social and political** issues, **like the exploitation of single**" | « *sense of character and* » ; $L_{\text{overlap}} = 24$ | 18.40 |
| $\theta'_3$ (ref) | OUT | "**like many western action films , this thriller is too loud and thoroughly overbearing , but its heartfelt** concern about **north korea 's recent past and south korea** 's future, its **sophisticated sense of character and its** penetrating views **on many social and political** issues, **like the exploitation of single**" | « *sophisticated sense of* » ; $L_{\text{overlap}} = 24$ | 18.30 |
| $\theta'_4$ (ref) | OUT | "**like many** western **action films , this thriller is too loud and thoroughly overbearing , but its heartfelt concern** about north korea **'s recent past and south** korea 's future, its sophisticated **sense of character and its** penetrating views **on many social and political** issues, **like the exploitation of** single" | « *sense of character and* » ; $L_{\text{overlap}} = 24$ | 17.93 |
| $\theta$ (target) | IN | "**like many** western **action films , this thriller is too loud and thoroughly overbearing , but its heartfelt concern about north korea 's recent past and south korea 's** future, its sophisticated **sense of character and its** penetrating **views on many social and political** issues, **like the exploitation of** single" | « *sense of character and* » ; $L_{\text{overlap}} = 24$ | 17.65 |

Table 10: Interpretability of the best MIA (2-gram) based on synthetic data for specialized canaries with $F = 30$, $\mathcal{P}_{\text{target}} = 31$ and $n_{\text{rep}} = 16$ for SST-2 from Figure 1(c). Results across 4 reference models and the target model for the canary with the **largest RMIA score** (most confidently and correctly identified as member by the MIA). Words in bold appear in 2-grams in $\widetilde{D}'$. The largest generated sub-sequence of the canary in $\widetilde{D}'$ corresponds to the maximum overlapping sub-string, not the longest sequence of words in bold.

| Model | IN or OUT | Canary (words present as part of 2-grams in $\widetilde{D}'$ in bold) | Max overlapping sub-string | 2-gram loss |
|---|---|---|---|---|
| $\theta'_1$ (ref) | IN | "**the star who helped give a spark to " chasing amy " and " changing lanes " falls flat as thinking man** cia **agent jack ryan in this summer 's big-budget** action drama, **" the hunt for red october** " (1990). **At the** time, bullet **time was used to** prolong" | « *the hunt for red october* » ; $L_{\text{overlap}} = 26$ | 18.12 |
| $\theta'_2$ (ref) | IN | "**the star who helped give a spark to " chasing amy " and " changing lanes " falls flat as thinking man** cia agent **jack ryan in this summer 's big-budget action** drama, **" the** hunt for red october " (1990). **At the** time, bullet **time was used to** prolong" | « *" and " changing lanes "* » ; $L_{\text{overlap}} = 29$ | 18.41 |
| $\theta'_3$ (ref) | OUT | "**the star who helped give a spark to " chasing amy " and "** changing lanes " **falls flat as thinking man** cia agent jack ryan **in this summer 's** big-budget action drama, **" the** hunt for red october " (1990). **At the** time, bullet **time was used to** prolong" | « *" chasing amy "* » ; $L_{\text{overlap}} = 19$ | 19.04 |
| $\theta'_4$ (ref) | OUT | "**the star** who helped **give a spark to " chasing amy " and " changing lanes " falls flat as thinking man** cia agent jack ryan **in this summer 's big-budget action** drama, **" the** hunt **for red** october " (1990). **At the** time, bullet **time was used to** prolong" | « *" and " changing lanes "* » ; $L_{\text{overlap}} = 29$ | 18.29 |
| $\theta$ (target) | OUT | "**the star who** helped **give a spark to " chasing amy " and "** changing lanes " **falls flat as thinking man** cia agent jack ryan **in this summer 's** big-budget action drama, **" the** hunt for red october " (1990). **At the** time, bullet **time was used to** prolong" | « *" chasing amy "* » ; $L_{\text{overlap}} = 19$ | 18.85 |

Table 11: Interpretability of the best MIA (2-gram) based on synthetic data for specialized canaries with $F = 30$, $\mathcal{P}_{\text{target}} = 31$ and $n_{\text{rep}} = 16$ for SST-2 from Figure 1(c). Results across 4 reference models and the target model for the canary with the **smallest RMIA score** (most confidently and correctly identified as non-member by the MIA). Words in bold appear in 2-grams in $\widetilde{D}'$. The largest generated sub-sequence of the canary in $\widetilde{D}'$ corresponds to the maximum overlapping sub-string, not the longest sequence of words in bold.

