# OpenReview forum: "The Canary’s Echo: Auditing Privacy Risks of LLM-Generated Synthetic Text"
_ICLR.cc/2025/Conference — Submitted to ICLR 2025_

### Official Review · Reviewer_oZg5 · 2024-10-25

**Soundness:** 3
**Presentation:** 3
**Contribution:** 3
**Rating:** 6
**Confidence:** 2

**Summary:**

This paper studies membership inference attacks (MIAs) that target data used to fine-tune pre-trained LLMs, which are used to generate synthetic data. The authors focus on the case where the adversary does not have access to the fine-tuned model, but only to the generated synthetic data. In this case, prior MIAs are not applicable because even though canaries may be memorized by the target model, they are less likely to be present in the synthetic data generated by in-distribution prompts. The authors address this by constructing canaries starting with an in-distribution prefix and transitioning into an out-of-distribution suffix, where the former increases the likelihood of influencing the synthetic data, and the latter increases the likelihood that the model memorizes them.

**Strengths:**

- The authors propose a new method for generating canaries optimized for this application.
- They run a thorough experimental evaluation to understand how the performance varies with the different parameters.

**Weaknesses:**

- It would be great if the authors could discuss concrete applications of this work or scenarios where it is useful.
- This work only gives a lower bound on the privacy risks, and it is not clear how the results generalize.

**Questions:**

- How would the results change if the adversary is able to choose the prompts for the synthetic data generation?
- How much hyperparameter tuning was done when training the model?

---

> ### Author Response · Authors · 2024-11-21
> **Rebuttal**
>
> We thank the reviewer for their feedback. We provide responses for the weaknesses (W) and questions (Q) raised below.
>
> > (W1) It would be great if the authors could discuss concrete applications of this work or scenarios where it is useful.
>
> Our work would be valuable to empirically evaluate the privacy risk associated with synthetic text data, for instance before releasing synthetic data for downstream analysis (e.g. synthetic summaries of medical consultations, synthetic customer support chats). In particular, MIA-based privacy auditing allows us to obtain empirical measures of residual privacy risks, detect bugs in implementations with formal privacy guarantees, incorrect assumptions, etc, to make an informed decision on whether the synthetic data is sufficiently anonymized. We will expand on the practical applications in the introduction.
>
> > (W2) This work only gives a lower bound on the privacy risks, and it is not clear how the results generalize.
>
> We agree with the reviewer that MIA-based privacy auditing only gives a lower bound on the privacy risk. We argue that this complements nicely with methods to synthesize data with differential privacy (DP) guarantees, which provide an upper bound on privacy risks. Auditing can not only detect violations of DP guarantees [1,2,3]  but also allows people to make less conservative decisions based on the privacy risks inferred from state-of-the-art attacks rather than purely on the upper bounds given by DP guarantees [4,5]. Indeed, DP upper bounds tend to be overly conservative because they consider unrealistic threat models (not only access to the fine-tuned model, but ability to adaptively choose intermediate model updates in DP-SGD).
>
> > (Q1) How would the results change if the adversary is able to choose the prompts for the synthetic data generation?
>
> In case the adversary can inject any arbitrary prompt, they can try to reconstruct training data (i.e. optimize for an input prompt that maximally reveals the presence of a data point - see Ye et al.[6]). This is a different threat model than what we assume in this paper (where the adversary only observes the released dataset).
>
> > (Q2) How much hyperparameter tuning was done when training the model?
>
> Before running the experiments, we ran a grid-search of the key finetuning hyperparameters (learning rate, batch size), picking the set resulting in the lowest validation loss. Based on these results (see figure below), we opt for a learning rate of 2e-5 and an effective batch size of 128. We will further expand on this in Sec. 7.
>
> **References**
>
> [1] Bichsel, B., Steffen, S., Bogunovic, I., & Vechev, M. (2021, May). Dp-sniper: Black-box discovery of differential privacy violations using classifiers. In 2021 IEEE Symposium on Security and Privacy (SP) (pp. 391-409). IEEE.
>
> [2] Stadler, T., Oprisanu, B., & Troncoso, C. (2022). Synthetic data–anonymisation groundhog day. In 31st USENIX Security Symposium (USENIX Security 22) (pp. 1451-1468).
>
> [3] Annamalai, M. S. M. S., Ganev, G., & De Cristofaro, E. (2024). " What do you want from theory alone?" Experimenting with Tight Auditing of Differentially Private Synthetic Data Generation (USENIX Security 2024).
>
> [4] Jagielski, M., Ullman, J., & Oprea, A. (2020). Auditing differentially private machine learning: How private is private SGD?. Advances in Neural Information Processing Systems, 33, 22205-22216.
>
> [5] Nasr, M., Songi, S., Thakurta, A., Papernot, N., & Carlin, N. (2021, May). Adversary instantiation: Lower bounds for differentially private machine learning. In 2021 IEEE Symposium on security and privacy (SP) (pp. 866-882). IEEE.
>
> [6] Ye, J., Borovykh, A., Hayou, S., & Shokri, R. Leave-one-out Distinguishability in Machine Learning. In The Twelfth International Conference on Learning Representations.

---

> > ### Comment · Reviewer_oZg5 · 2024-11-25
> >
> > Thank you for the clarifications, it would be great if you could expand on the practical applications in the introduction. I will keep my score.

---

### Official Review · Reviewer_d3yN · 2024-11-01

**Soundness:** 3
**Presentation:** 3
**Contribution:** 2
**Rating:** 5
**Confidence:** 3

**Summary:**

The paper investigates privacy auditing of synthetically generated text using MIAs. The authors demonstrate that standard canaries for model-based MIAs are ineffective for data-based MIAs. Hence, they design a new canary generation technique that improves data-based MIAs. The main challenge of generating effective canaries is ensuring that they align with the prompt, so they can at least be generated. But at the same time need to be unique enough so that they are memorized by the model. So they create a canary where the prefix contains a truncated original record, and the suffix is generated by a pre-trained LLM.

**Strengths:**

1. The paper tackles an important problem of privacy auditing synthetic text generation from LLMs
2. The ideas in the paper are easy to follow
3. The result showing that increased canary perplexity leads to lower MIA AUC for synthetic data-based attacks is interesting

**Weaknesses:**

1. I find the main results from 5.1 that current canary designs aren’t effective for MIAs on synthetic data a bit underwhelming. I feel like this is already well-known and intuitive that MIAs based on just the generated labels/tokens is much harder than having logits because you loose the uncertainty quantities that are usually exploited in standard MIAs. Unless I am missing something that is the novelty of the results in 5.1.
2. It’s unclear to me how the author’s proposed canary generation method improves over the standard canary generation. For example, how much does the canaries with in-distribution suffix improve over the baseline canaries for MIAs on synthetic data in terms of AUC? This would help with connecting the main results from 5.1 as I only see improvements stated in terms of TPR which feels disconnected from 5.1.
3. There seems to be a TPR improvement for increasing prefix length on SST-2 in Figure 1, but marginal improvement for AG News. This ambiguity in results makes it inconclusive about the exact effect the prefix length has improving MIAs (large vs small). Unfortunately, we’d need to see more results on at least one more dataset.
4. I guess generalizng points 1. and 2., I’m uncertain about the exact contribution of the paper. It is hard to find in the introduction and there is no conclusion section to make the contribution more concrete.

**Questions:**

1. Are the membership scores using n-gram probability and similarity metric novel techniques that you are proposing?
2. The authors mention that memorization is more likely when there is an abrupt transition in entropy between sub-sequences. Could you discuss the intuition behind this?
3. Were experiments on low-perplexity prefix lengths of 40 and 50 performed? It would be great to provide those results, too.

---

> ### Author Response · Authors · 2024-11-21
> **Rebuttal (part 1)**
>
> We thank the reviewer for their feedback. We provide responses for the weaknesses (W) and questions (Q) raised below.
>
> > (W1)  I feel like this is already well-known and intuitive that MIAs based on just the generated labels/tokens is much harder than having logits because you loose the uncertainty quantities that are usually exploited in standard MIAs
>
> While the results may be intuitive, we are -to our knowledge- the first to propose a methodology to measure leakage of training data through release of synthetic text data and quantify the gap compared to model-based attacks. Moreover, the fact that MIAs based on synthetic data results in worse performance than the model-based MIA, does not answer questions about what type of memorized data is leaked and what properties of that data influence the leakage. With our proposed methods, we further explore which canary construction methods are more optimal to audit the privacy of synthetic text data.
>
> Further, while there has been research on MIAs against synthetic data in the tabular and image domain, we argue that the notion of model-based attack in these cases are sufficiently distinct or even not well-defined. For instance, Hayes et al. and Hilprecht et al. develop data-based MIAs for synthetic images generated by GANs, for which a model-based attack could use the GAN’s discriminator, which we argue is quite different from a generative LLM. Further, the notion of a model-based attack is not necessarily well defined for other use-cases such as many statistical ways to compute synthetic tabular data considered by Stadler et al.
>
> > (W2) It’s unclear to me how the author’s proposed canary generation method improves over the standard canary generation.
>
> First, we would like to clarify that prior work has only considered model-based MIAs to audit the privacy of LLMs, consistently considering high-perplexity sequences as canaries. Canaries were first introduced by Carlini et al. (2019) as a sequence containing random numbers (mimicking e.g. a credit card number), later also adopted by for instance Stock et al. (2022). More recently, Wei et al. (2024) considers sequences of random tokens while Meeus et al. (2024b) confirms that high perplexity, synthetically generated canaries are more vulnerable.
>
> In this paper, we find that currently deployed canary generation techniques fall short for MIAs based on synthetic data - a contribution we argue is novel and important in itself. We show that in-distribution canaries, while highly suboptimal canaries for model-based attacks, are much better canaries for data-based MIAs.
>
> We then explore how canaries with in-distribution prefixes and out-of-distribution suffixes could improve upon this. To study this in more detail, we will add the results for in-distribution canaries in Figures 1c and 1f. We will also create this figure for a third dataset. We estimate these experiments to be finished by next Monday.
>
> > (W3) Unfortunately, we’d need to see more results on at least one more dataset.
>
> We agree that our findings would carry more weight when also confirmed for a third dataset. We are running this for another dataset and will add the results in the revised manuscript as part of the rebuttal (anticipated by next Monday).
>
> > (W4)  I’m uncertain about the exact contribution of the paper.
>
> Our main objective is to investigate the factors that influence the leakage of memorized training data when synthetic data is generated by LLMs.
> - While numerous empirical studies have explored **memorization** in LLMs, these primarily focus on the models themselves, assuming that the MIA adversary has direct access to the models for generating data or computing probabilities.
> - However, similar investigations are lacking in scenarios where LLMs are used to generate synthetic data.
> - In this context, we study what types of memorized data are susceptible to leakage, what properties of the data contribute to this leakage, and how these factors interplay. Specifically, we consider properties such as the number of repetitions, data length, perplexity, and whether the data is in-distribution or out-of-distribution.
>
> In order to perform such analysis, we:
>
> 1. Introduce MIA techniques enabling data-based attacks against synthetic text data.
> 2. Quantify the gap between model- and data-based MIA performance for synthetic data attacks.
> 3. Demonstrate that canaries as traditionally used for model-based attacks against LLMs fail when auditing privacy risks of synthetic text data. Instead, we find that in-distribution canaries work better than high perplexity canaries and we explore a better canary design by leveraging in-distribution canary prefixes and out-of-distribution, synthetically generated suffixes.
>
> We will make this contribution more clear in the introduction.

---

> > ### Author Response · Authors · 2024-11-21
> > **Rebuttal (part 2)**
> >
> > > (Q1) Are the membership scores using n-gram probability and similarity metric novel techniques that you are proposing?
> >
> > To our knowledge, we are the first to consider n-gram probability and similarity based metrics to compute MIA scores for synthetic text data. On top of that, we also adapt the computation of RMIA scores as originally proposed by Zarifzadeh et al. (2024) to our setup (see Appendix B).
> >
> > We do note that for computing these MIA scores, we draw inspiration from prior work on MIAs against synthetic data from the tabular and image domain. For instance, prior work has considered training models on the synthetic data (e.g. new discriminator on synthetic images from GANs (Hayes et al., 2019)) or using distance-based metrics to infer membership (Hilprecht et al., 2019; Yale et al., 2019; Hyeong et al., 2022; Zhang et al., 2022). While we adopt similar techniques from prior work, we are the first to expand (and evaluate) these techniques to the domain of synthetic text data.
> >
> > > (Q2) The authors mention that memorization is more likely when there is an abrupt transition in entropy between sub-sequences. Could you discuss the intuition behind this?
> >
> > Thank you for this question. Rereading our manuscript, we find that the results we currently have do not allow us to draw any conclusion about the use of an  "abrupt" versus “smooth” transition in perplexity of canaries. We will adapt this to "memorization is more likely when canaries contain sub-sequences with high perplexity".
> > > (Q3) Were experiments on low-perplexity prefix lengths of 40 and 50 performed? It would be great to provide those results, too.
> >
> > We like to clarify that for this canary generation, we take a certain, unmodified prefix from an in-distribution sample and generate a synthetic suffix. We apply rejection sampling to only retain the canaries with a certain target perplexity for the entire sequence.
> >
> > When the prefix length becomes as high as 40 (40 out of 50 words are fixed), it becomes very hard to generate suffixes (of length 10 words) bringing the overall canary to the target perplexity. Hence, including results for F=40 has not been possible. We will, however, include the results for F=50 (full in-distribution canary) in figures 1(c,f).

---

> > > ### Comment · Reviewer_d3yN · 2024-11-29
> > > **Response by Reviewer d3yN**
> > >
> > > I thank the authors for their response. They have addressed most of my concerns, and hence, I have raised my score accordingly. However, I await their experimental results on an additional dataset to address my last concern about the exact effectiveness of the prefix length.

---

### Official Review · Reviewer_uo6v · 2024-11-04

**Soundness:** 4
**Presentation:** 3
**Contribution:** 3
**Rating:** 8
**Confidence:** 4

**Summary:**

This paper investigates membership inference attacks (MIAs) on synthetic text generated by fine-tuned Large Language Models (LLMs). It highlights that traditional MIAs, which rely on access to the model or its logits, are not directly applicable to scenarios where only the synthetic data is released.  The authors find that standard "canary" examples, designed to be highly vulnerable to model-based MIAs, are less effective when auditing the privacy risks of synthetic data because their influence on the generated text diminishes.  To address this, they propose crafting specialized canaries with an in-distribution prefix and an out-of-distribution suffix to improve their impact on synthetic data generation while maintaining memorizability. Their experiments demonstrate that this approach significantly enhances the power of MIAs on synthetic text, enabling a more accurate assessment of the privacy risks associated with releasing LLM-generated data. They also analyze the membership inference signal and find that it relies on subtle shifts in the word probability distribution within the synthetic data, rather than explicit regurgitation of canary subsequences.

**Strengths:**

* Makes progress on an important problem by proposing new methods to perform MIA against synthetic data instead of directly with model access, and developing a sound methodology to design canaries for this task

* Thorough empirical evaluation providing quanlitative and quantitative analysis of the different aspects that affect MIA success

**Weaknesses:**

* Lacks a centralized presentation of the overall auditing strategy. In particular, Algorithm 1 is presented as adding a single canary one time to the dataset, while in practice most of the evaluations add several canaries with multiple repetitions. The authors should considering spelling out the full auditing method with all hyper-parameters and design choices in a single algorithm block.

* The description of the threat model (L128-138) only focuses on the access mode available to the adversary. This needs to be expanded to capture the knowledge of the training pipeline available to the adversary (base dataset, training method, sampling method, etc). This is important because in some cases (e.g. when using shadow models and when designing canaries for synthetic data auditing) the adversary leverages some of this information to perform the attack, while in other methods the attacker uses less information. A table summarizing these differences would make it easier to compare the different approaches evaluated in the paper.

**Questions:**

See weaknesses for points that I would like to see addressed in the rebuttal.

---

> ### Author Response · Authors · 2024-11-21
> **Rebuttal**
>
> We thank the reviewer for their feedback. We provide responses for the weaknesses (W) raised below.
>
> > (W1) The authors should considering spelling out the full auditing method with all hyper-parameters and design choices in a single algorithm block.
>
> We agree with the reviewer that the notion of repetitions for canaries is not specified in the formalization in Algorithm 1. We will add this to the revised manuscript.
>
> > (W2) The description of the threat model (L128-138) only focuses on the access mode available to the adversary. This needs to be expanded to capture the knowledge of the training pipeline available to the adversary (base dataset, training method, sampling method, etc)
>
> We agree with the reviewer that not all assumptions made for the attacker are made explicit in the threat model section. We will further clarify this in the manuscript, as exemplified in the table below:
>
> | Assumptions made for the adversary                                            | Model-based MIA | Data-based MIA |
> |-------------------------------------------------------------------------------|-----------------|----------------|
> | Knowledge of the full training dataset D used to train the target model (apart from the knowledge of included canaries). | x               | x              |
> | Knowledge of the training procedure of the target model.                      | x               | x              |
> | Knowledge of the prompt template p used to generate the synthetic data.       | x               | x              |
> | Query-access to the target model, returning predicted logits.                 | x               |                |
> | Knowledge of the strategy employed to sample synthetic data from the target model (temperature, top-k). |                 | x              |
>
> Where we further clarify:
> - In both threat models, we consider the attacker to be able to train reference models. This always assumes access to the original training dataset, and the exact training procedure of the target model.
> - The model-based MIA uses the prompt template to compute the model loss for the target record, as specified in 3.1.1. The data-based MIA uses the prompt template to generate synthetic data.
> - Only the model-based attack has query-access to the target model, able to retrieve token-level, predicted logits for any input sequence.
> - Only the data-based attack also generates synthetic data as part of the reference models, so only this threat model leverages the sampling hyperparameters.

---

### Official Review · Reviewer_yLLE · 2024-11-06

**Soundness:** 3
**Presentation:** 3
**Contribution:** 2
**Rating:** 5
**Confidence:** 4

**Summary:**

The paper investigates the use of MIAs to audit privacy in synthetically generated text. The authors have designed new canaries that begin with an in-distribution prefix (to maximize the chance of regeneration by the fine-tuned model) and an out-of-distribution suffix to maximize memorization.

**Strengths:**

- The problem is intriguing, particularly as designing MIAs becomes challenging without access to the target model.
- The paper is well-written.

**Weaknesses:**

- In Table 1, the authors use only 30 words for in-distribution canaries, compared to 50 words for synthetic canaries (as mentioned in a footnote). A larger canary size generally results in higher AUC-ROC numbers. Even for SST-2, the in-distribution canaries perform better than synthetic canaries, which diminishes the contribution of the authors regarding the efficiency of their canaries. This pattern is also visible in Tables 6 and 7.
- The AUC-ROC scores for model attacks in Table 1 suggest that the fine-tuned model is overfitted (likely due to a high n_rep=12). Knowing this, results of MIA scores for synthetic data are little bit low. The scores using SIM_jac and SIM_emb are nearly random, with only the 2-gram model scores slightly better. It also raises the question of the practicality of having high n_rep values like 12 or 16 in real-world synthetic data privacy auditing. These results do not strongly support the new MIA contribution. Additionally, Figure 1 (a and b) show near-random curves for n_rep=2 or 4.

**Questions:**

- It would be helpful to see the MIA scores when transitioning from the last words of the prefix to the first words of the suffix in generated canaries to demonstrate their effectiveness.
- It would benefit the paper if the authors could highlight their contributions more effectively based on the results.
- How costly is it to compute the n-gram model when dealing with a large volume of synthetic data generated by the target model? A cost analysis for different n-grams and data sizes would be informative.
- Given a large corpus of synthetic data, would it be feasible to train a transformer-based attack model to predict the next word probabilities as the MIA scores?

---

> ### Author Response · Authors · 2024-11-21
> **Rebuttal (part 1)**
>
> We thank the reviewer for their feedback. We provide responses for the weaknesses (W) and questions (Q) raised below.
>
> > (W1) Even for SST-2, the in-distribution canaries perform better than synthetic canaries, which diminishes the contribution of the authors regarding the efficiency of their canaries.
>
> Foremost, we argue that finding that high perplexity canaries are insufficient for privacy auditing of synthetic text data is a novel and valuable contribution in itself. The privacy auditing of LLMs currently explored in prior work exclusively considers the model-based attack, which has consistently considered high perplexity sequences as canaries. In this paper, we show that currently used canary generation techniques fail for MIAs based on synthetic data. We show that in-distribution canaries, while highly suboptimal canaries for model-based attacks, are much better canaries for data-based MIAs.
>
> We then explore how canaries with in-distribution prefixes and out-of-distribution suffixes could improve upon this. To study this in more detail, we will add the results for in-distribution canaries in Figures 1c and 1f. We will also create this figure for a third dataset. We estimate these experiments to be finished by next Monday.
>
> When it comes to the results in Table 1, we agree that there is a difference in length between in-distribution canaries (30 words) and synthetic canaries (50 words), making a fair comparison harder. As expressed in the footnote, we do this because there are not enough samples in SST-2 with 50 or more words. Would it be useful if we would run the same set of experiments for SST-2 with synthetic canaries of length 30?
>
> > (W2) The AUC-ROC scores for model attacks in Table 1 suggest that the fine-tuned model is overfitted (likely due to a high n_rep=12). Knowing this, results of MIA scores for synthetic data are little bit low… It also raises the question of the practicality of having high n_rep values like 12 or 16 in real-world synthetic data privacy auditing
>
> While we agree that the target model is likely overfitted to the canaries in the scenario of Table 1 (due to n_rep >10), we argue this is by design to optimally study privacy auditing of synthetic data. We investigate the gap between model- and data-based attacks under similar conditions in an auditing scenario. We are particularly interested in measuring the worst-case attack performance which better highlights privacy violations.
>
> Specifically, the scenario in Table 1 largely advantages model-based attacks (in-distribution prefix F = 0, canary perplexity = 250), and so data-based attacks necessitate a large number of repetitions (n_rep = 12) to significantly beat a random baseline. The goal of presenting Table 1 is exactly to illustrate that data-based attacks perform poorly compared to model-based attacks in a typical auditing scenario with canaries with high perplexity.
>
> Figure 1 (c) presents a fairer comparison using canaries of moderate perplexity and adapted to each attack (F = 0 for model-based attacks and F > 0 for data-based attacks). Still, and as expected, we observe a gap between model- and data- based attacks. We reiterate that our goal is not to close this gap (and indeed, we give reasons why we believe it is inherent), but to quantify and explain why it exists.
>
> Lastly, we argue that the canary repetitions do not impact the utility of the synthetic data. The downstream performance remains highly similar for synthetic data generated with or without canaries (see App D, Table 4,5).
>
> > (W2) These results do not strongly support the new MIA contribution
>
> To our knowledge, we are the first to propose and evaluate various ways to compute MIA scores for synthetic text data. On top of that, we adapt the computation of RMIA scores as originally proposed by Zarifzadeh et al. (2024) to our setup (see Appendix B). These methods allow us to (i) reach data-based MIA performances significantly larger than a random guess and (ii) quantify the gap in MIA performance between model- and data-based attacks. Together, our MIA contribution allows for the first empirical quantification of privacy risk associated with synthetic text data.
>
> While we believe these MIA methodologies are novel, our primary claim is not that these attacks achieve high performance in practice. Rather, we introduce an initial toolkit for privacy auditing of synthetic text data, quantify the gap with model-based attacks and examine optimal canary design within this framework.
>
> > (Q1) It would be helpful to see the MIA scores when transitioning from the last words of the prefix to the first words of the suffix in generated canaries to demonstrate their effectiveness.
>
> Thank you for the suggestion. We will add results on MIA scores for varying length of the target sequence, indicating the transition from prefix to suffix, and will include these results as part of the rebuttal (anticipated by next Monday).

---

> > ### Author Response · Authors · 2024-11-21
> > **Rebuttal (part 2)**
> >
> > >  (Q2) It would benefit the paper if the authors could highlight their contributions more effectively based on the results.
> >
> > Our main objective is to investigate the factors that influence the leakage of memorized training data when synthetic data is generated by LLMs.
> > - While numerous empirical studies have explored **memorization** in LLMs, these primarily focus on the models themselves, assuming that the MIA adversary has direct access to the models for generating data or computing probabilities.
> > - However, similar investigations are lacking in scenarios where LLMs are used to generate synthetic data.
> > - In this context, we study what types of memorized data are susceptible to leakage, what properties of the data contribute to this leakage, and how these factors interplay. Specifically, we consider properties such as the number of repetitions, data length, perplexity, and whether the data is in-distribution or out-of-distribution.
> >
> > In order to perform such analysis, we:
> > 1. Introduce MIA techniques enabling data-based attacks against synthetic text data.
> > 2. Quantify the gap between model- and data-based MIA performance for synthetic data attacks.
> > 3. Demonstrate that canaries as traditionally used for model-based attacks against LLMs fail when auditing privacy risks of synthetic text data. Instead, we find that in-distribution canaries work better than high perplexity canaries and we explore a better canary design by leveraging in-distribution canary prefixes and out-of-distribution, synthetically generated suffixes.
> >
> > We will make our contribution more clear in the introduction.
> >
> > > (Q3) How costly is it to compute the n-gram model when dealing with a large volume of synthetic data generated by the target model? A cost analysis for different n-grams and data sizes would be informative.
> >
> > Fitting the n-gram model on the generated synthetic data is extremely cheap, especially compared to training the reference models. For instance, for both SST-2 (43k samples) and AG-News (120k samples), training the n-gram model on the synthetic data takes less than 1 CPU-minute.
> >
> > > (Q4) Given a large corpus of synthetic data, would it be feasible to train a transformer-based attack model to predict the next word probabilities as the MIA scores?
> >
> > Thank you for this suggestion. We agree that it would be interesting to train a more sophisticated model on the synthetic data instead of the n-gram model. We will finetune a small GPT-2-based model on the synthetic data and will include the resulting data-based MIA performance (using the finetuned GPT-2’s loss instead of the n-gram’s loss) as a new column in Table 1 (anticipated by next Monday).

---

> > > ### Comment · Reviewer_yLLE · 2024-12-02
> > >
> > > While I appreciate the authors' efforts, my largest weaknesses regarding performance of their method (for low n_rep) and novelty of the paper still persist. I hence keep my score.

---

### Author Response · Authors · 2024-11-28
**Revised manuscript as part of the rebuttal**

We have revised the manuscript to address comments from the reviewers:
- We added results for in-distribution canaries to Figure 1(c,f) and Figure 6 (c,f). We find that canaries with an in-distribution prefix and an out-of-distribution suffix consistently lead to better MIA performance (in terms of higher AUC ROC score, but especially in terms of higher TPR at low FPR) than in-distribution or high perplexity OOD canaries normally used in model-based attacks (addressing comments by Reviewers yLLE and d3yN).
- We have made explicit and clearer our contributions in the introduction (Reviewers yLLE and d3yN).
- In the introduction, we have elaborated on real-world applications and on how our work complements upper bounds on privacy risks when synthetic data is generated using models fine-tuned with differential privacy guarantees (Reviewer oZg5).
- We have modified Algorithm 1 to account for the number of canary repetitions and make it closer to the auditing method we use in experiments. We have also added an explicit parameter standing for the decoding strategy used to generate synthetic data (Reviewer uo6v).
- We have summarized the capabilities of adversaries in the data- and model-based threat models and what capabilities the attacks in our experiments effectively use in Appendix D, complementing the description in lines 128-138 and the formalization in Algorithm 1 (Reviewer uo6v).
- We added in Appendix J a description of hyperparameter tuning for fine-tuning LLMs (Reviewer oZg5).

We have also made numerous minor changes throughout the manuscript, improving the presentation, notational consistency and clarity. We moved Section 5.3 to Appendix K.1 to accommodate for the new content without compromising legibility.

Due to contingency on compute resources, we are still working on getting additional experimental results on an additional dataset (Reviewer d3yN) and leveraging a Transformer model instead of an $n$-gram model in attacks (Reviewer yLLE). We will elaborate on our progress in the following days as part of the discussion and will work diligently to include the results in an eventual camera-ready version of the paper.

---

> ### Author Response · Authors · 2024-12-04
> **Reply to revised manuscript as part of the rebuttal**
>
> Due to technical reasons, we have not been able to generate results for an additional, third dataset in time for the end of the rebuttal phase. Yet, we hope that the reviewers and AC recognize that our results are consistent for the two datasets already considered in the manuscript, supporting the claims made in the introduction.
>
> In particular, we find that canaries with in-distribution prefix and an out-of-distribution suffix consistently lead to better MIA performance (in terms of higher AUC ROC score, but especially in terms of higher TPR at low FPR) than in-distribution or high perplexity OOD canaries normally used in model-based attacks (see Figure 1 (c,f) and Figure 6 (c,f)). We will work diligently to include results for an additional dataset in an eventual camera-ready version of the paper.

---

### Meta-Review · Area_Chair_pNw3 · 2024-12-21

**Metareview:**

The submission studies membership inference attacks (MIAs) that target data that is used to fine-tune pre-trained LLMs which are then used to synthesize data. The submission focuses on the setting where the adversary does not have access to the fine-tuned LLM, but only has access to the synthetic data corpus.The submission shows that using canaries designed to maximize the vulnerability to attacks that have access to the model are sub-optimal when auditing privacy risks in the setting where only the synthetic data is released.
While the submission studies an important problem, and presents an interesting approach, a concern was raised in the reviews about the empirical evaluation being limited. The authors attempted to include the evaluation on an additional dataset during the discussion period, but didn’t manage to do so. We think that adding such an additional evaluation would be needed before the paper is ready for publication.

**Additional Comments On Reviewer Discussion:**

The reviews raised a few different concerns about this submission including:

1) The experimental evaluation
2) The presentation of the auditing strategy
3) The description of the threat model
4) Clarifying the paper’s contribution
5) Clarifying concrete applications of the work

While the authors have mostly addressed concerns 2), 3), 4) and 5) satisfactorily, the concern around 1) remains, in particular in terms of including an evaluation on an additional dataset.

---

### Decision · Program_Chairs · 2025-01-22

Reject